# CONTRAST WITH AGGREGATION:
# A SCALABLE FRAMEWORK FOR MULTI-VIEW REPRESENTATION LEARNING

## ABSTRACT

Multi-View Representation Learning (MVRL) aims to learn the joint representation from diverse data sources by discovering complex relationships among them. In MVRL, since the downstream task information and the view availability are often unknown a-priori, it is essential for the joint representation to be robust to the partial availability of views. However, existing methods exhibit various limitations, such as discarding potentially valuable view-specific information, lacking the ability to extract representation from an arbitrary subset of views, or requiring considerable computational resources that increase exponentially with the number of views. To address these challenges, we present a scalable MVRL framework based on contrastive learning. Our approach employs a set of encoders that is able to extract representations from arbitrary subset of views, and jointly trains them with a computation cost that scales linearly with the number of views. We conducted comprehensive evaluations across 7 MVRL benchmark datasets ranging from 2 to 8 views, demonstrating that our method robustly handles diverse input view combinations and outperforms strong baseline methods.

## 1 INTRODUCTION

Multi-View Representation Learning (MVRL) focuses on learning the joint representation of instances from various types of views without relying on label or task information (Hwang et al., 2021). By uncovering the potentially complex relationships among views, the joint representation must capture important underlying factors from these views, which is essential for downstream tasks such as data fusion equipped with multiple sensors (Zhang et al., 2011) and medical diagnoses based on diverse records (Yuan et al., 2018; Zhang et al., 2018).

Providing more views of each instance usually helps eliciting more accurate representations, but it poses three major challenges. Firstly, it requires more sophisticated cross-view association, which involves identifying shared and view-specific factors of variation across all views, under the varying levels of correlations among views. Secondly, increasing the number of views typically increases the difficulty and cost of collecting the data, necessitating the ability to handle missing views during learning and inference stages. Lastly, it significantly raises the computational cost of learning the representation. For example, to address the scenario where an arbitrary set of views is missing, we could simply learn the representations for every subset of views, but this typically results in computation costs that grow exponentially with the number of views, rendering the approach unscalable.

Recent MVRL approaches leveraging Contrastive Learning (CL) (Tian et al., 2020; Poklukar et al., 2022) or VAEs (Wu & Goodman, 2018; Shi et al., 2019; Sutter et al., 2021; Hwang et al., 2021) have shown promising results in downstream tasks (e.g., classification) where capturing shared information across views is critical. However, these approaches typically have at least one of the following limitations: (1) discarding view-specific information that could be relevant to downstream tasks (Tian et al., 2020; Poklukar et al., 2022), (2) lacking a mechanism to learn representations from any subset of views (Wu & Goodman, 2018; Shi et al., 2019; Tian et al., 2020; Poklukar et al., 2022; Hwang et al., 2021), or (3) incurring computational cost that grows exponentially (Sutter et al., 2021) or quadratically (Tian et al., 2020) with the number of views.

In this work, we propose a scalable, information-theoretic MVRL framework that effectively addresses these three challenges. First, we formulate MVRL as the problem of encoding representations from every subset of views that are informative enough to capture both view-specific and shared factors of variation. We then introduce an information-theoretic objective that jointly trains all subset-view representations with a computational cost that scales linearly with the number of views: by combining the Mahalanobis distance and the InfoNCE (Oord et al., 2018; Poole et al., 2019) objective, we derive a variational lower bound that calibrates each representation to generalize well in downstream tasks. Through extensive evaluations on 7 MVRL benchmark datasets, we demonstrate that our method robustly encodes representations from various combinations of input views and outperforms strong baseline methods. Our contributions are three-fold:

1. **Theoretical contribution**: Proposition 1 in Section 3.2 formally shows that the single Mutual Information (MI) term between complete views and the joint representation encoded by Mixture of Experts (Shi et al., 2019) or its variants lower bounds the weighted average of various MI terms. This enables efficient representation learning for all subsets of views.

2. **Algorithmic contribution**: In Section 3.3, we derive a tractable lower bound for our single MI objective, allowing the MoPoE (Sutter et al., 2021) joint encoder to learn and calibrate exponentially many subset-view representations with a computational cost that scales linearly with the number of views. Importantly, this represents a significant improvement over prior work (Sutter et al., 2021), which trained the same encoder with a computational cost that increased exponentially with the number of views.

3. **Empirical contribution**: By conducting comprehensive evaluations on 7 MVRL benchmark datasets spanning 2 to 8 views, we demonstrate the robustness of our method across diverse input-view combinations, consistently surpassing strong baseline methods.

## 2 RELATED WORK

**Multi-View Fusion**    MVRL methods can be categorized into early fusion and late fusion approaches (Liang et al., 2021), depending on how they encode multiple views. Early fusion methods encode all views into a joint representation by feeding a stack of input views to one joint encoder. For instance, transformer models are trained with learning objectives such as masked reconstruction (He et al., 2022; Geng et al., 2022; Georgescu et al., 2022; Shi et al., 2022; Mo & Morgado, 2023) or with autoregressive modeling (Ramesh et al., 2021; Wang et al., 2022b; 2023; Wu et al., 2024) on multiple input views. Although these methods benefit from high expressivity in encoding the joint representation, they commonly suffer from heavy computational costs that scale quadratically with the number of views. On the other hand, late fusion methods encode each view into a representation with a dedicated encoder per view and then aggregate these single-view representations into one joint representation. Contrastive Multi-View Representation Learning methods (Tian et al., 2020; Poklukar et al., 2022; Radford et al., 2021; Cherti et al., 2023) and Multi-View VAEs (Wu & Goodman, 2018; Shi et al., 2019; Sutter et al., 2020; 2021; Hwang et al., 2021) fall into this category. These late fusion approaches are closely related to our work and are further reviewed below.

**Contrastive Multi-View Representation Learning**    Contrastive Multi-View Coding (CMC)(Tian et al., 2020) is one of the most representative works in Contrastive MVRL and has been applied to pretraining multimodal foundation models (Radford et al., 2021; Cherti et al., 2023). It optimizes the InfoNCE (Oord et al., 2018; Poole et al., 2019) objective between every pair of single-view representations from different views by maximizing their cosine similarity, thereby aligning representations from multiple views. However, its single-view representations are encouraged to capture only the shared factors of variation since its InfoNCE terms are upper-bounded by the mutual information (MI) between two views (Cover, 1999; Wang et al., 2022a), which quantifies the amount of shared information. Furthermore, the computational cost combinatorially increases with the number of views, making it difficult to apply CMC to a large number of views. In contrast, GMC (Poklukar et al., 2022) employs a complete-view representation to align the single-view representations by maximizing the cosine similarity between the complete-view representation and each single-view representation. Although its computational cost scales linearly with the number of views, it also suffers from discarding view-specific factors due to maximizing the cosine similarities, as observed in our experiments (Sec 4.1). Additionally, GMC employs two backbone encoders for each view, doubling the number of encoder parameters.

**Multi-View VAEs** Multi-View VAEs (Wu & Goodman, 2018; Shi et al., 2019; Sutter et al., 2021; Hwang et al., 2021) learn the joint representation with multiple single-view VAEs by maximizing the evidence lower bound (ELBO). MVAE (Wu & Goodman, 2018) employs the Product of Experts (PoE) as its joint encoder, which effectively aggregates information from all views using an Inverse-Variance Weighted (IVW) average of single-view representations. Since PoE struggles with calibrating each single-view encoder, MVTCAE (Hwang et al., 2021) derives Conditional Variational Information Bottlenecks (CVIBs) from their Total Correlation objective to calibrate single-view encoders toward the PoE joint encoder. In contrast, MMVAE (Shi et al., 2019) employs the Mixture of Experts (MoE) as its joint encoder, which takes an arithmetic mean of single-view encoders. Although MoE explicitly optimizes each single-view encoder, it potentially fails to aggregate information from multiple views, just taking each view-specific representation as a separate component in the mixture. To improve MMVAE, MoPoE-VAE (Sutter et al., 2021) introduces a Mixture of Product of Experts (MoPoE) as its joint encoder, which defines the representation of each subset of views by the PoE of views in the subset and combines all subset-view representations using MoE. Although this approach significantly improves MMVAE, it requires computing the density from the MoPoE, resulting in a computational cost that grows exponentially with the number of views. We further investigate these encoder structures in Section 3.1.

Since our method is also a late-fusion approach that trains the MoPoE joint encoder with a contrastive learning objective, it is closely related to both Contrastive MVRL methods and Multi-View VAEs. A detailed comparison between these methods and our approach is provided in Section B.

## 3 METHOD

Let $N$ be the total number of views under consideration and $v_i$ be the $i$-th view, $1 \leq i \leq N$. In addition, let $v_{1:N} = \{v_i\}_{i=1}^N$ denote a complete-view data instance drawn from an unknown data distribution $p_D(v_{1:N})$ and $v_s$ denote any non-empty subset of $v_{1:N}$ such that $v_s \subseteq v_{1:N}$. For example, if $N = 3$, then $v_s \in \{v_1, v_2, v_3, v_{12}, v_{13}, v_{23}, v_{123}\}$, where $v_{12} = \{v_1, v_2\}$, $v_{13} = \{v_1, v_3\}$, $v_{23} = \{v_2, v_3\}$, and $v_{123} = \{v_1, v_2, v_3\}$. Additionally, let $\theta_s$ and $z_s$ be the parameter of the stochastic encoder and the encoded representation of $v_s$, e.g., $z_{23} \sim p_{\theta_{23}}(\cdot \mid v_{23})$.

Our objective is to learn an informative representation $z_s$ for every subset of views by capturing all factors of variation within the input views $v_s$. To achieve this, we maximize the Mutual Information (MI) between $z_s$ and $v_s$ for each subset of views as shown below:

$$\sum_{V_s \subseteq V_{1:N}}^{N} I_{\theta_s}(Z_s; V_s) \overset{(N=3)}{=} I_{\theta_1}(Z_1; V_1) + I_{\theta_2}(Z_2; V_2) + I_{\theta_3}(Z_3; V_3) \tag{1}$$

$$+ I_{\theta_{12}}(Z_{12}; V_{12}) + I_{\theta_{13}}(Z_{13}; V_{13}) + I_{\theta_{23}}(Z_{23}; V_{23}) + I_{\theta_{123}}(Z_{123}; V_{123})$$

Maximizing equation 1 requires the representation of each combination of views to be informative to its input views. This allows each $z_s$ to capture not only the shared factors of variations but also view-specific ones.

It is important to note that equation 1 differs from $\sum_{1 \leq i < j \leq N} I_{\theta_i, \theta_j}(Z_i; Z_j)$, the sum of MI between every pair of single-view representations optimized by CMC (Tian et al., 2020); each MI term $I_{\theta_i, \theta_j}(Z_i; Z_j)$ in CMC encourages learning only the shared factors of variation, since it is upper-bounded by $I(V_1; V_2)$. A more detailed comparison between our method and CMC can be found in Section B.1 of the supplementary material.

However, direct optimization of equation 1 presents two challenges:

1. **Scalability** Equation 1 costs a large amount of computation due to the number of (1) encoder parameters and (2) objective terms increase exponentially with the number of views.

2. **Calibration** Each subset-view representation is optimized independently, leading to inconsistent subset-view representations for subsets derived from the same data instance.

To address these issues, we propose a scalable subset-view representation learning framework based on Contrastive Learning (CL). We start by reviewing existing encoder structures to reduce the number of encoder parameters (Sec. 3.1). Then, we show that a single-term objective function allows us

to learn every subset-view representation at the computation cost that grows linearly with the number of views (Sec. 3.2). Finally, we introduce a tractable lower bound of our objective that effectively calibrates subset-view representations based on CL (Sec. 3.3).

## 3.1 PARAMETER SHARING AMONG SUBSET-VIEW ENCODERS

Since it is not scalable to employ independent encoders for every combination of views, we consider the late fusion approach. Specifically, we encode each single view $v_i$ with a Gaussian distribution such that $p_{\theta_i}(z_i|v_i) = N(\mu_i, \sigma_i^2)$ for $1 \leq i \leq N$ and combine any set of views by Product of Experts (PoE) (Hinton, 2002) which is also known as Inverse-Variance Weighted (IVW) average (Cochran & Carroll, 1953; Cochran, 1954).

$$p_{\theta_s}(z_s|v_s) \triangleq N\left(\mu_s, \sigma_s^2 \mathbf{I}\right), \quad \text{where} \quad \mu_s \triangleq \frac{\sum_{v_i \in v_s} \mu_i/\sigma_i^2}{\sum_{v_i \in v_s} 1/\sigma_i^2} \quad \text{and} \quad \sigma_s^2 \triangleq \frac{1}{\sum_{v_i \in v_s} 1/\sigma_i^2}. \quad (2)$$

Earlier works have shown that single-view encoders must be calibrated to be effectively aggregated from a set of views by PoE (Hwang et al., 2021; 2023), which we address by calibrating all subset-view encoders including single-view ones in Section 3.3. Additional discussion on the statistical properties and optimality of IVW within the context of MVRL can be found in Section G.

When these exponentially many encoders are all combined using a Weighted Mixture of Experts (WMoE[1]), the number of parameters of the joint encoder is linearly proportional to the number of views. This results in the joint encoder structured as follows:

$$p_\theta(z \mid v_{1:N}) = \sum_{v_s \subseteq v_{1:N}} \lambda_s \cdot p_{\theta_s}(z_s|v_s), \text{ where } \theta = \{\theta_i\}_{i=1}^N \text{ and } 0 \leq \lambda_s \leq 1, \sum_{v_s \subseteq v_{1:N}} \lambda_s = 1. \quad (3)$$

$$\overset{(N=3)}{=} \lambda_1 p_{\theta_1}(z_1|v_1) + \lambda_2 p_{\theta_2}(z_2|v_2) + \lambda_3 p_{\theta_3}(z_3|v_3)$$
$$+ \lambda_{12} p_{\theta_{12}}(z_{12}|v_{12}) + \lambda_{13} p_{\theta_{13}}(z_{13}|v_{13}) + \lambda_{23} p_{\theta_{23}}(z_{23}|v_{23}) + \lambda_{123} p_{\theta_{123}}(z_{123}|v_{123}).$$

Assigning $\lambda_s = \frac{1}{N}$ to single-view encoders and zeros on the rest reduces WMoE to the typical MoE (Shi et al., 2019), assigning $\lambda_s = 1$ only on complete-view encoder reduces WMoE to PoE (Wu & Goodman, 2018; Hwang et al., 2021). In addition, evenly distributing $\lambda_s = \frac{1}{2^N-1}$ to all encoders yields MoPoE (Sutter et al., 2021), the mixture of all subset-view encoders. Although assigning different values of $\lambda_s$ can encourage WMoE to focus on some subsets of views during training, we do not consider any sophisticated assignment scheme in our work as we do not assume prior knowledge of downstream tasks or their view availabilities.

Although MoPoE-VAE (Sutter et al., 2021) jointly learns all subset-view representations using the MoPoE joint encoder, it requires computing the density of its joint encoder, resulting in exponentially many computations of densities of all subset-view encoders. In contrast, we train the MoPoE joint encoder only at the computation cost that *linearly* scales with the number of views, which we will discuss in the following sections.

## 3.2 SCALABLE SUBSET-VIEW REPRESENTATION LEARNING WITH A SINGLE TERM

In addition to reducing the number of encoder parameters, we need to reduce the computations as well. A direct optimization of exponentially many terms in equation 1 is not desirable. Instead, we derive that any WMoE encoder that maximizes the single MI term $I_\theta(Z; V_{1:N})$ can jointly train all its subset-view encoders.

**Proposition 1.** *Given the WMoE joint encoder $p_\theta$ defined as equation 3, $I_\theta(Z; V_{1:N}) \leq \sum_{v_s \subseteq v_{1:N}} \lambda_s \cdot I_{\theta_s}(Z_s; V_s)$, i.e. $I_\theta(Z; V_{1:N})$ lower bounds the weighted average version of equation 1.*

*Proof.* See Section A in the supplementary material. □

---

[1]We refer to the joint encoder in equation 2 as WMoE to distinguish it from the equally-weighted sum of experts, which is commonly referred to as Mixture of Experts (MoE) in existing literature (Shi et al., 2019; Sutter et al., 2021; Hwang et al., 2021).

Proposition 1 indicates that maximizing $I_\theta(Z; V_{1:N})$ between the joint representation and the complete views jointly maximizes multiple $I_{\theta_s}(Z_s; V_s)$ terms, each defined between a subset-view representation and its input views. Consequently, the MoPoE joint encoder can maximize $\frac{1}{2^N-1} \sum_{v_s \subseteq v_{1:N}} I_{\theta_s}(Z_s; V_s)$, which aligns with our goal. It is also notable that MoE joint encoder would maximize $\frac{1}{N} \sum_{i=1}^N I_{\theta_i}(Z_i; V_i)$, while the PoE joint encoder would maximize $I_{\theta_{1:N}}(Z_{1:N}; V_{1:N})$. We analyzed the impact of the encoder choice in Section E.7.

Although reducing exponentially many MI terms to one MI term is beneficial, direct computation of $I_\theta(Z; V_{1:N}) = \mathbb{E}_{p_D(v_{1:N})} [D_{KL} [p_\theta(\boldsymbol{z} \mid v_{1:N}) || p_\theta(\boldsymbol{z})]]$ is intractable because computing the density of $p_\theta(z) = \int p_D(v_{1:N}) p_\theta(z \mid v_{1:N}) dv_{1:N}$ is involved with the unknown density $p_D$.

To resolve this issue, we can maximize any of the sample-based MI estimators (Belghazi et al., 2018; Hjelm et al., 2018; Oord et al., 2018; Poole et al., 2019) that lower bound $I_\theta(Z; V_{1:N})$.

### 3.3 CALIBRATING SUBSET-VIEW REPRESENTATIONS WITH CL

We maximize our MI objective $I_\theta(Z; V_{1:N})$ with InfoNCE (Oord et al., 2018) due to its low-variance MI estimation and numerical stability (Poole et al., 2019). Since InfoNCE is also utilized by CMC (Tian et al., 2020) to maximize $\sum_{1 \leq i < j \leq N} I_{\theta_i, \theta_j}(Z_i; Z_j)$, adopting it allows for a direct comparison between CMC and ours to isolate the effect of optimizing different MI objectives. Given our joint encoder, $I_\theta(Z; V_{1:N})$ can be lower-bounded by the following InfoNCE objective.

$$I_\theta(Z; V_{1:N}) \geq \hat{I}_\theta^{NCE}(Z; V_{1:N}) \triangleq \mathbb{E}_{\prod_{k=1}^K p_D(v_{1:N}^{(k)}) p_\theta(z^{(k)} | v_{1:N}^{(k)})} \left[ \frac{1}{K} \sum_{i=1}^K \log \frac{e^{f\left(z^{(i)}, v_{1:N}^{(i)}\right)}}{\frac{1}{K} \sum_{j=1}^K e^{f\left(z^{(i)}, v_{1:N}^{(j)}\right)}} \right], \tag{4}$$

where $K$ is the minibatch size and $f$ is a learnable critic function that helps tighten the bound. To align representations from different views, similarity measures for $f$, such as Cosine similarity (Tian et al., 2020; Poklukar et al., 2022; Radford et al., 2021; Cherti et al., 2023) and Euclidean distance (Wang et al., 2022a), have been widely applied to the InfoNCE objective to enhance generalization in downstream tasks with varying view availability. However, these measures are not well-suited for our multivariate Gaussian representations: Cosine similarity is inapplicable as our representations are not L2-normalized, and Euclidean distance is less ideal because it assumes uniform scaling across all dimensions, which does not align with the properties of our representations.

Instead, we define $f$ as the *Mahalanobis* distance between the joint representation $z$ and $p_{\theta_{1:N}}(z_{1:N} | v_{1:N}) = N(\mu_{1:N}, \sigma_{1:N}^2 \mathbf{I})$ as shown below.

$$f(z, v_{1:N}) = -\frac{(z - \mu_{1:N})^T \sigma_{1:N}^{-2} \mathbf{I}(z - \mu_{1:N})}{\tau}, \tag{5}$$

where $\tau$ is the temperature. Here, $f$ encodes $v_{1:N}$ into $\mu_{1:N}, \sigma_{1:N}^2$, which determine the distribution of the complete-view representation $z_{1:N}$. These parameters are then used to calibrate $z$ by enforcing it to infer $z_{1:N}$. As $z$ is sampled from one of subset-view encoders $p_{\theta_s}(z_s | v_s)$, randomly selected by the WMoE joint encoder with probability $\lambda_s$, this effectively calibrate $z_s$.

Substituting equation 5 into $\hat{I}_\theta^{NCE}(Z; V_{1:N})$ encourages the positive pair $(z^{(i)}, v_{1:N}^{(i)})$ to be closer in the representation space, while pushing the negative pair $(z^{(i)}, v_{1:N}^{(j)})$ further apart. This process naturally enforces $z^{(i)}$ to aggregate all factors of variation underlying its input $v_s^{(i)}$ as these factors are also present in $v_{1:N}^{(i)}$, and thus utilizing them makes it easier to colocate the positive pair and dislocate the negative pair. The optimization process for $\hat{I}_\theta^{NCE}(Z; V_{1:N})$ is dipicted in Figure 1.

Finally, we apply a Variational Information Bottleneck (VIB) (Alemi et al., 2017) to each single-view encoder to prevent overfitting to the training data, yielding the final objective function:

$$\hat{I}_\theta^{NCE}(Z; V_{1:N}) - \beta \sum_{i=1}^N \lambda_i \cdot D_{KL} [p_{\theta_i}(z_i | v_i) || N(0, \mathbf{I})], \tag{6}$$

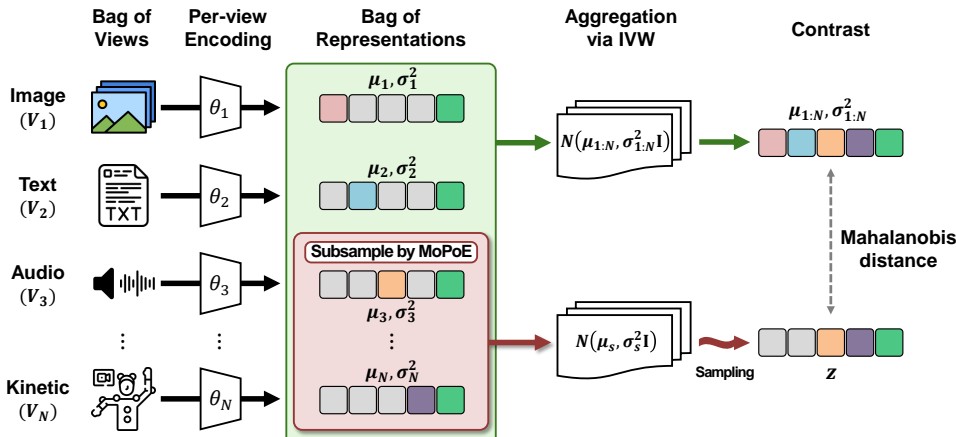

Figure 1: The optimization process of the InfoNCE objective (equation 4). Each view is encoded separately into $\mu_i, \sigma_i^2$, representing each single-view encoder $p_{\theta_i}(z_i|v_i)$. Subsequently, $\{\mu_i, \sigma_i^2\}_{i=1}^N$ are aggregated into $\mu_{1:N}, \sigma_{1:N}^2$ via IVW, which form complete-view encoder $p_{\theta_{1:N}}(z_{1:N}|v_{1:N})$ (green arrow). In addition, a subset of $\{\mu_i, \sigma_i^2\}_{i=1}^N$, randomly selected by MoPoE, is aggregated into $\mu_s, \sigma_s^2$ to sample the joint representation $z$ from the selected subset-view encoder $p_{\theta_s}(z_s|v_s)$ (red arrow). Finally, the Mahalanobis distance between $z$ and $\langle \mu_{1:N}, \sigma_{1:N}^2 \rangle$ is computed to optimize the InfoNCE objective, which jointly learns the subset-view and complete-view representations.

where $\beta$ is a hyperparameter that controls the magnitude of regularization for each view's encoder. We call our method as Contrast with Aggregation (CwA), which trains the MoPoE encoder by maximizing equation 6. CwA explicitly learns to encode from every subset of input views with a computational cost that scales linearly with the number of views.

Unlike MoPoE-VAE, CwA bypasses the density computation of the joint MoPoE encoder, resulting in an overall computation complexity of $O(N)$. Specifically, the optimization of equation 6 requires the density computations of (1) $N$ single-view encoders, (2) the complete-view encoder, (3) and one subset-view encoder uniform-randomly chosen by the MoPoE joint encoder. Algorithm 1 outlines each step of training CwA and its associated computation costs in terms of the number of views, which can be found in Section F.

## 4 EXPERIMENTS

To evaluate the quality of the representation learned by our method, we conducted evaluations in linear regression and linear classification tasks in three sets of experiments. In all experiments, we aim to see if our method can robustly perform given any subset of all views.

**Baseline methods**  We compared our method with strong baseline methods including CMC (Tian et al., 2020), GMC (Poklukar et al., 2022), MoPoE-VAE (Sutter et al., 2021), and MVTCAE (Hwang et al., 2021). Brief reviews of these baseline methods can be found in Section 2. Since CMC lacks a joint representation of multiple views, we computed the average of single-view representations to aggregate multiple views. Similarly, since GMC learns to aggregate only complete views, we computed the average of single-view representations when subset-view representations are available in the downstream tasks. In addition to these methods, we included GMCs, a variant of GMC that has only one backbone encoder per view, similar to other comparing methods. All results are averaged over 10 independent runs. Detailed information on the hyperparameter settings of each method can be found in Section D.

### 4.1 8 VIEWS FROM SYNTHETIC DATASET

To evaluate whether our method can infer both view-specific and shared factors of variation while scaling to many views, we generated a synthetic dataset composed of 8 views. For each instance, 2 types of data-generative factors are sampled: a view-specific factor $g_i \sim [0, 2]$ for each view ($1 \le i \le 8$) and a shared factor $g_s \sim [-1, 1]$. Then, each view $v_i \in \mathbb{R}^{100}$ is generated by drawing

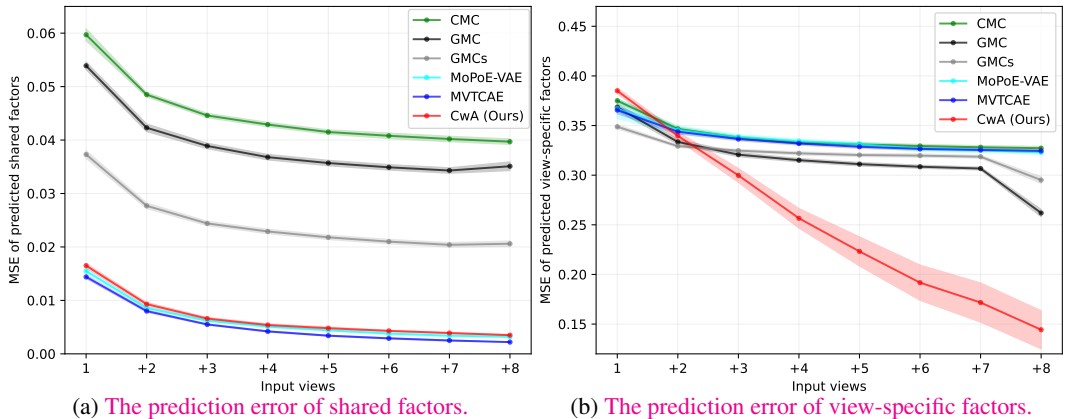

(a) The prediction error of shared factors.   (b) The prediction error of view-specific factors.

Figure 2: Results of linear regression on Synthetic dataset. Mean squared error between true data-generative factors and predicted factors is measured with incrementally adding views.

100 samples from the Gaussian distribution $N(g_s, g_i^2)$, resulting in vectorized views. Every $v_i$ is encoded by its dedicated MLP encoder of each method. We generated 10,000 instances and split them into train(8), valid(1), test(1) sets where the values in the parentheses represent the data split ratio. Further details about the experiment, including network architectures and visualization of the data generation process, can be found in Section D.1 of the supplementary material.

While capturing sample mean and variance in each view helps discover all data-generative factors, observing many views should improve the identification of $g_s$, the common mean of all views.

**Evaluation protocol**    We trained linear regression models to predict shared and view-specific data-generative factors using the frozen representation. Specifically, we pretrained each method using the train set for 1,000 epochs, validating every 10 epochs. During validation, we trained a linear regression model with z of the complete views, each to predict the true data-generative factors $[g_1; ...; g_8; g_s]$ in the train set. We then evaluated it using z of the complete views in the validation set. We saved the regression model and each method when their performance was the best in the validation set. After training, we evaluated the saved models by measuring 2 different Mean Squared Errors (MSE): one between true shared generative factor $g_s$ and that predicted from z of accumulated input views (e.g. view 1, views 1+2, ..., views 1:N) in the test set, and the other between true view-specific generative factors $[g_1; ...; g_8]$ and those predicted from the same z.

**Results**    Figure 2 compares the MSE for predicting the shared factor $g_s$ (Figure 2a) and the view-specific factors $[g_1; ...; g_8]$ (Figure 2b). Due to the space limitation, the MSE for jointly predicting the shared and view-specific factors $[g_1; ...; g_8; g_s]$ is presented in Section E.7. The x-axis represents the input view(s) accumulated one by one, and the y-axis indicates the MSE. While CwA exhibits a slightly higher error in predicting view-specific factors compared to other methods when using 1∼2 views, it effectively reduces its prediction error for both shared and view-specific factors as more views are added. Specifically, it significantly reduces the error of view-specific factors, demonstrating that CwA identifies the view-specific factor of each view and effectively aggregates this factor in its representation. This is due to the optimization of our main objective function $I_\theta(Z; V_{1:N})$, which maximizes $I_{\theta_s}(Z_s; V_s)$ for all subset views $v_s$ and their representation $z_s$. This ensures that every $z_s$ captures all factors of variation including view-specific ones across $v_s$. This property is further supported by the tractable lower bound of our MI objective in equation 4, which encourages $z_s$ to aggregate information across its input views, making it easier to classify positive and negative pairs in the representation space as we discussed in Section 3.3. As a result, CwA internally computes not only the shared mean across views but also the variance of each view with simple MLP encoders.

Conversely, the other methods generally fail to leverage additional views when aggregating view-specific factors. CMC is guided by its MI objective function $\sum_{1 \leq i < j \leq N} I_{\theta_i, \theta_j}(Z_i; Z_j)$, which emphasizes capturing only shared factors between views, so additional views help only in identifying the shared factors. Comparing CwA with CMC highlights the importance of carefully selecting the MI terms to optimize. GMC(s) focuses on aligning views using cosine similarity maximization between complete-view and each single view representations, resulting in representations that primarily capture shared factors while neglecting view-specific ones. Lastly, due to their reliance on

reconstructing input views, MoPoE-VAE and MVTCAE capture sampling noise incurred by view-specific variances $g_i^2$ in the data generation process rather than discovering true view-specific factors. This leads to poorer performance in aggregating view-specific factors.

**Runtime statistics**  To evaluate the scalability of our method, we measured the running time of each representation learning method. Figure 3 shows the result. The x-axis represents the total number of views used for training and the y-axis represents the total amount of time for running 10 training epochs. The result shows that the running time of both GMCs and CwA inrceases linearly with the number of views, demonstrating the least amount of time. This is because they both have computational costs that grow linearly with the number of views and use one encoder for each view. While GMC and MVTCAE also have linear computational costs, they are relatively slower because GMC additionally employs one additional encoder for each view and MVTCAE uses a decoder, doubling the size of their models. Conversely, CMC shows a significant increase in running time with the number of views

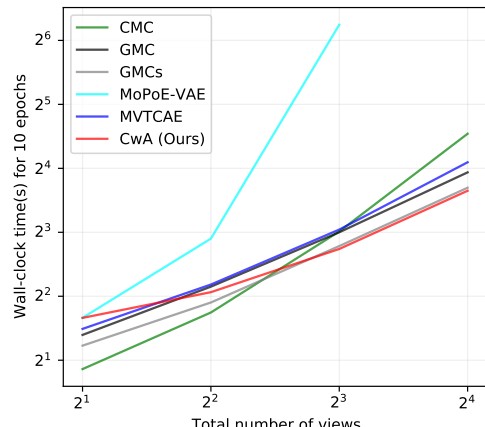

Figure 3:  Running time of training each method for 10 epochs.

due to the combinatorial pairwise comparisons required by its contrastive learning objective, making it less scalable. Lastly, MoPoE-VAE shows a significant increase in computational cost as the number of input views increases due to the density computation of each expert in MoPoE as discussed in Section 3.1. Remarkably, CwA can be trained much faster than MoPoE-VAE, despite also using the MoPoE joint encoder.

## 4.2 Video, Audio, and Text views

To evaluate the ability of our method to extract meaningful information from realistic multi-view data, we utilized MultiBench (Liang et al., 2021), a well-established collection of real-world multi-modal datasets. Specifically, we selected four datasets from MultiBench: MOSI (Zadeh et al., 2016), MUSTARD (Castro et al., 2019), FUNNY (Hasan et al., 2019), and MOSEI (Zadeh et al., 2018). These datasets were designed to explore human affective states through diverse expressions, including spoken language, facial expressions, gestures, and speech tone. Representing human expressions as multimodal time-series data across text, video, and audio modalities (views), these datasets enable tasks such as predicting sentiment (MOSI), emotion (MOSEI), humor (FUNNY), and sarcasm (MUSTARD). The complementary nature of these views highlights the importance of understanding their intricate relationships. Additional details about these datasets are provided in Section D.2.

**Evaluation protocol**  We trained a linear classifier to predict sentiment (MOSI), sarcasm (MUS-TARD), humor (FUNNY), and emotions (MOSEI) for each dataset. We pretrained each method using the train set for 1,000 epochs, validating every 10 epochs. During validation, we trained a lin-ear classifier using z of the complete views in the validation set and evaluated the classifier with z of the complete views in the validation set. We saved the classifier and representation learning models when their performance was the best in the validation set. After training, we evaluated the saved models by measuring the classification accuracy predicted from z of all input view combinations.

**Results**  Table 1 presents the results. Each column reports the classification accuracy for each input view combination, except the 6th and 10th columns, which show the average performances for 1 and 2 views, respectively. The **best** performance is written in bold, while the 2nd best performance is underlined in each column. Due to space limitations, we present the standard error in Section D.2.

When a single view is given, although CwA shows the best average performance in MOSI, MUS-TARD, and FUNNY datasets, it underperforms in several cases compared to the best-performing method in each dataset, such as GMC and GMCs in MOSEI. However, when 2 views are jointly given, our method outperforms GMC and GMCs in most cases, resulting in the best average perfor-mance in all four datasets. This is because GMC and GMCs are limited to optimizing only single-view and complete-view representations in their formulations, while CwA calibrates all subset-view representations, allowing better utilization of any subset composed of multiple views.

| Dataset | Method | 1 view | | | | 2 views | | | | 3 views |
| --- | --- | --- | --- | --- | --- | --- | --- | --- | --- | --- |
| | | Video | Audio | Text | Avg. | V,A | V,T | A,T | Avg. | V,A,T |
| MOSI | CMC | 54.11 | 52.76 | 62.41 | 56.42 | 54.15 | 60.92 | 59.46 | 58.18 | 57.76 |
| | GMC | 52.83 | 54.14 | _62.67_ | 56.55 | 54.52 | 60.48 | 60.04 | 58.35 | 62.78 |
| | GMCs | 52.43 | _55.6_ | 62.55 | 56.86 | _55.54_ | 60.09 | 61.47 | 59.03 | _62.99_ |
| | MoPoE-VAE | _54.3_ | **56.66** | 60.87 | _57.28_ | **56.25** | 59.1 | 61.63 | 58.99 | 59.48 |
| | MVTCAE | **54.81** | 54.23 | 62.24 | 57.09 | 55.31 | _62.99_ | _63.82_ | _60.7_ | 62.94 |
| | CwA (Ours) | 53.85 | 54.68 | **67.23** | **58.59** | 54.68 | **66.18** | **67.52** | **62.79** | **65.51** |
| MUSTARD | CMC | _58.04_ | 57.1 | _63.7_ | _59.61_ | 58.26 | **64.13** | **64.28** | _62.22_ | _63.91_ |
| | GMC | 55.72 | 57.54 | **64.49** | 59.25 | 58.12 | 62.83 | 63.91 | 61.62 | 60.22 |
| | GMCs | 57.83 | _57.54_ | 62.25 | 59.2 | _58.7_ | 62.39 | 61.88 | 60.99 | 62.25 |
| | MoPoE-VAE | 49.71 | 51.45 | 53.19 | 51.45 | 55.72 | 51.67 | 51.67 | 53.02 | 57.25 |
| | MVTCAE | 51.3 | 49.2 | 54.49 | 51.67 | 47.17 | 52.1 | 50.51 | 49.93 | 56.74 |
| | CwA (Ours) | **59.28** | **57.75** | 63.48 | **60.17** | **60.65** | _63.99_ | _64.13_ | **62.92** | **64.28** |
| FUNNY | CMC | 55.18 | 57.33 | 59.66 | 57.39 | 58.83 | 59.23 | 61.8 | 59.95 | 62.57 |
| | GMC | 54.13 | 57.35 | **60.9** | 57.46 | 58.54 | 61.31 | **62.65** | 60.83 | _63.23_ |
| | GMCs | 54.67 | **58.45** | 59.62 | _57.58_ | _59.4_ | 60.04 | 61.92 | 60.45 | 61.89 |
| | MoPoE-VAE | 52.74 | 57.01 | 58.13 | 55.96 | 58.93 | 60.22 | 61.26 | 60.14 | 62.43 |
| | MVTCAE | _55.91_ | 56.1 | 59.94 | 57.32 | 59.14 | **62.2** | 62.0 | _61.12_ | 62.68 |
| | CwA (Ours) | **56.08** | _57.48_ | _60.26_ | **57.94** | **60.02** | _61.67_ | 62.58 | _61.42_ | **63.36** |
| MOSEI | CMC | 66.67 | 70.39 | 75.76 | 70.94 | 70.66 | 74.87 | 75.15 | 73.56 | 74.48 |
| | GMC | _69.1_ | _70.81_ | **76.05** | _71.99_ | _71.08_ | _75.43_ | _75.9_ | _74.14_ | _76.26_ |
| | GMCs | **69.31** | **70.82** | _75.84_ | _71.99_ | **71.11** | 75.25 | 75.41 | 73.92 | 75.95 |
| | MoPoE-VAE | 57.7 | 53.37 | 56.06 | 55.71 | 55.33 | 58.88 | 69.94 | 61.38 | 70.96 |
| | MVTCAE | 61.62 | 58.59 | 64.61 | 61.61 | 59.26 | 69.04 | 54.0 | 60.77 | 70.85 |
| | CwA (Ours) | 68.34 | 70.75 | 73.95 | 71.01 | 70.83 | **76.81** | **76.91** | **74.85** | **77.53** |

Table 1: Classification accuracy (%) of the learned representation of subset views in MultiBench.

With all three views combined, our method outperforms all competing methods across all datasets, demonstrating its ability to effectively aggregate information from multiple views. This observation can be clearly seen in Table 2, which reports the number of times each method performed the best for each number of views.

Although CMC shows competitive performance in the MUSTARD dataset, it underperforms in the other three datasets compared to ours, especially in multiple view scenarios. This is because CMC does not learn to aggregate information from multiple views, focusing only on pair-wise optimizations of single-view representations. Lastly, compared to CL methods, VAE methods commonly underperform in most cases due to the high dimensionality of

| Method | 1 view (16 cases) | 2 views (16 cases) | 3 views (4 cases) | Total (36 cases) |
| --- | --- | --- | --- | --- |
| CMC | 0 (3) | 2 (1) | 0 (1) | 2 (5) |
| GMC | 3 (4) | 1 (4) | 0 (2) | 4 (10) |
| GMCs | 4 (4) | 1 (3) | 0 (1) | 5 (8) |
| MoPoE-VAE | 1 (2) | 1 (0) | 0 (0) | 2 (2) |
| MVTCAE | 1 (1) | 1 (4) | 0 (0) | 2 (5) |
| CwA (Ours) | **7** (2) | **10** (4) | **4** (0) | **21** (6) |

Table 2: The number of performing the best (2nd best) in each number of input views in MultiBench.

video, audio, and text views, which imposes difficulty in discovering their relationships. As a result, reconstructing views from the representation leads to memorizing views rather than discovering the underlying factors of variation.

We observe certain cases where adding more views results in decreased performance across all methods. For example, the text view alone achieves the highest performance for CMC on MOSI and MOSEI, GMC on MUSTARD, and CwA on MOSI. Similarly, MoPoE and MVTCAE fail to enhance the performance of the text view when video or audio is added as additional views on MUSTARD. This phenomenon arises because the informativeness of views is highly unbalanced for the downstream task. Specifically, the text view is inherently more informative for sentiment inference, as it often includes keywords that make the task straightforward. This explains why the text view consistently outperforms other single views across all methods.

In such scenarios, combining representations from multiple views through a (weighted) average may slightly degrade the representation from the most informative view. This occurs because each view's representation contributes to all dimensions of the combined representation, potentially diluting the signal from the dominant view.

### 4.3    6 VIEWS UNDER A COMPLEX CORRELATION

To evaluate if our method can effectively aggregate information across many views under a complex correlation, we assessed our method on a multi-view dataset generated by Li et al. (2015). The dataset comprises 6 visual features including Histogram of Oriented Gradients (Dalal & Triggs, 2005), GIST (Oliva & Torralba, 2001), and Local Binary Pattern (Ojala et al., 2002). These visual features were extracted from images in the Caltech-101 (Fei-Fei et al., 2004) dataset and treated as independent views. We split the data into train(8), valid(1), test sets(1), where the values in parentheses represent the split ratio. Detailed information on data preprocessing, visual features, and network architectures can be found in Section D.3 of the supplementary material.

Learning the joint representation of these views is complex because they are generated from different types of lossy compressions, containing complementary information.

**Evaluation protocol**    We trained a linear classifier to predict the label of each instance same as in Section 4.2. In addition to all comparing methods, we also evaluated CwA+recon, which is CwA that additionally trains decoders that reconstruct views from the joint representation and minimizes reconstruction losses.

**Results**    Figure 4 summarizes the results. The x-axis represents the number of input views, and the y-axis represents the performance averaged over subsets with the same number of views. Unlike earlier experiments, VAE methods (MoPoE-VAE, MVTCAE, CwA+recon) generally outperform all CL methods including CwA. This is because the input views are features that the representation possibly needs to learn; reconstruction-based methods have a better chance of memorizing the input views by reducing reconstruction loss. Remarkably, CwA+recon that jointly optimizes reconstruction loss and our objective function equation 6 effectively improves MoPoE-VAE, outperforming all the comparing methods.

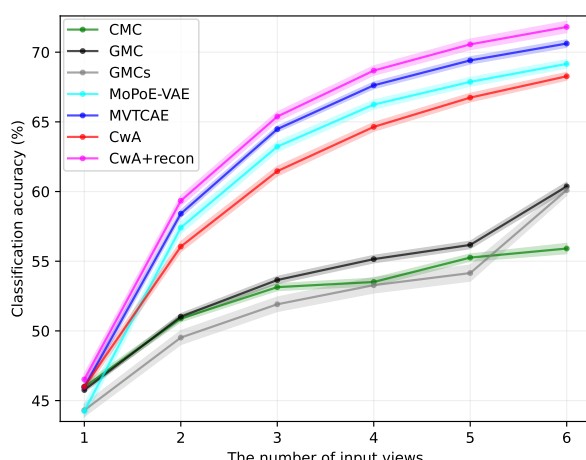

Figure 4: Classification results in Caltech101 dataset.

Showing the performance competitive to MoPoE-VAE, CwA considerably outperforms all CL methods, improving its performance with additional views. The result implies that CwA effectively optimizes each subset-view representation without discarding any meaningful information. On the other hand, GMC, GMCs, and CMC barely improve their performance with additional views. This is because they commonly align single-view representations using cosine similarity, encouraging them to be equal. As a result, although views are containing complementary information, the representation from each view tends to lose any view-specific factors potentially important to the task.

Due to space limitations, additional results including hyperparameter sensitivity analysis, visualization of learned representations, training on missing-view data, and evaluation on ImageNet (Deng et al., 2009) are provided in Section E.

## 5    CONCLUSION

In this work, we introduced Contrast with Aggregation (CwA), a scalable MVRL framework that effectively aggregates information from any subset of views. By formulating an information-theoretic objective applicable to existing encoder models, we enabled the optimization of every subset-view representation with a computational cost that increases linearly with the number of views. Additionally, by integrating the Mahalanobis distance into the InfoNCE objective, we reformulated our method as a CL approach that calibrates each subset-view representation. Extensive evaluations on synthetic and real-world datasets demonstrated CwA's superior robustness, significantly outperforming existing methods in leveraging additional views and aggregating information across different views.

## ETHICS STATEMENT

Our method can be applied in situations where multiple sensors in a multi-sensor system malfunction, potentially posing security risks. For example, in self-driving vehicles with multiple camera inputs, adverse weather conditions like rain can corrupt the sensors. Additionally, this method could decrease the number of views needed for the system to function, which might help reduce the overall carbon footprint. However, there is a potential misuse case where the reduction in the number of views could be exploited to compromise the system's security. For instance, if the system relies on fewer sensor inputs, it might become more vulnerable to targeted attacks that spoof or manipulate the limited data available. Such vulnerabilities could lead to scenarios where the self-driving vehicle misinterprets its surroundings, potentially causing accidents or unauthorized access to the vehicle's control systems. Therefore, while the method offers significant benefits, it is crucial to implement robust safeguards to prevent and mitigate any potential security threats arising from reduced sensor inputs.

## REPRODUCIBILITY STATEMENT

To ensure reproducibility, we provide a detailed evaluation protocol for all three sets of experiments, including the processes for training, validation, and testing, as described in Section 4. We also present comprehensive information on hyperparameters, network architectures, and data pre-processing in Section D.

Additionally, we have included an anonymized link to our code below, which contains three repositories: Syn (Sec 4.1), Multi (Sec 4.2), and Cal (Sec 4.3). Each repository includes a README file with instructions for reproducing the results presented in the paper.
`https://anonymous.4open.science/r/CwA_codes/`

We kindly request that reviewers download the code before everything including reviews are made public. We will reactive the link and make the code public when the paper is published.

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

# Supplementary Material

## CONTENTS

## A    PROOF

*Proof of Proposition 1.* Given the WMoE joint encoder $p_\theta$ defined as equation 3, $I_\theta(Z; V_{1:N}) \le \sum_{v_s \subseteq v_{1:N}} \lambda_s \cdot I_{\theta_s}(Z_s; V_s)$, i.e. $I_\theta(Z; V_{1:N})$ lower bounds the weighted average version of equation 1.

$$\underbrace{I_\theta(Z; V_{1:N})}_{\star} = \mathbb{E}_{p_D(v_{1:N})} \left[ D_{KL} \left[ p_\theta(\boldsymbol{z} \mid v_{1:N}) || p_\theta(\boldsymbol{z}) \right] \right] \tag{7}$$

$$= \mathbb{E}_{p_D(v_{1:N})} \left[ D_{KL} \left[ \sum_{v_s \subseteq v_{1:N}} \lambda_s \cdot p_{\theta_s}(z_s \mid v_s) || \sum_{v_s \subseteq v_{1:N}} \lambda_s \cdot p_{\theta_s}(z_s) \right] \right] \tag{8}$$

$$\le \mathbb{E}_{p_D(v_{1:N})} \left[ \sum_{v_s \subseteq v_{1:N}} \lambda_s \cdot D_{KL} \left[ p_{\theta_s}(z_s \mid v_s) || p_{\theta_s}(z_s) \right] \right] \tag{9}$$

$$= \sum_{v_s \subseteq v_{1:N}} \lambda_s \cdot \mathbb{E}_{p_D(v_s)} \left[ D_{KL} \left[ p_{\theta_s}(z_s \mid v_s) || p_{\theta_s}(z_s) \right] \right] = \sum_{v_s \subseteq v_{1:N}} \lambda_s \cdot \underbrace{I_{\theta_s}(Z_s; V_s)}_{\star\star}.$$

Equation 8 holds because the latter term in KL in equation 7 can be decomposed as $p_\theta(z) = \sum_{v_s \subseteq v_{1:N}} \lambda_s p_{\theta_s}(z_s)$, which we show in Proposition 2. The inequality in equation 9 holds due to the convexity of KL divergence. □

**Proposition 2.** *Given the WMoE joint encoder $p_\theta$ defined as Eq. equation 3, the marginal distribution of the joint representation $p_\theta(z)$ is the weighted mixture of $p_{\theta_s}(v_s)$, the marginal distributions of subset-view representations.*

*Proof.*

$$p_\theta(z) = \int p_D(v_{1:N}) p_\theta(z \mid v_{1:N}) dv_{1:N}$$

$$= \int p_D(v_{1:N}) \sum_{v_s \subseteq v_{1:N}} \lambda_s \cdot p_{\theta_s}(z_s \mid v_s) dv_{1:N}$$

$$= \sum_{v_s \subseteq v_{1:N}} \lambda_s \int p_D(v_s) p_{\theta_s}(\boldsymbol{z}_s \mid v_s) dv_s$$

$$= \sum_{v_s \subseteq v_{1:N}} \lambda_s \cdot p_{\theta_s}(z_s). \tag{10}$$

Although the direct computation of Eq. equation 10 is intractable due to the unknown density $p_D$, we can still observe that the marginal distribution $p_\theta(z)$ is also WMoE whose experts are the marginal distributions of subset-view representations. □

# B    DETAILED COMPARISON TO RELATED WORKS

| Methods | Computation Cost | Learning **Single-View** Representations | Learning **Subset-View** Representations | Learning **Complete-View** Representation | Decoder Free |
|---|---|---|---|---|---|
| CMC | $O(N^2)$ | O | X | X | O |
| GMC(s) | $O(N)$ | O | X | O | O |
| MVAE | $O(N)$ | X | X | O | X |
| MMVAE | $O(N)$ | O | X | X | X |
| MoPoE-VAE | $O(2^N)$ | O | O | O | X |
| MVTCAE | $O(N)$ | O | X | O | X |
| CwA (ours) | $O(N)$ | O | O | O | O |

Table 3: Quick comparison of various multi-view (multimodal) representation learning methods.

## B.1    CMC (TIAN ET AL., 2020)

By pairwise comparison between single-view representations of views, CMC trains single-view encoders. Specifically, it optimizes the following objective.

$$\sum_{1 \le i < j \le N} I_{\theta_i, \theta_j}(Z_i; Z_j) \overset{(N=3)}{=} I_{\theta_1 \theta_2}(Z_1; Z_2) + I_{\theta_1 \theta_3}(Z_1; Z_3) + I_{\theta_2 \theta_3}(Z_2; Z_3), \qquad (11)$$

where $z_1 \sim p_{\theta_1}(\cdot \mid v_1), z_2 \sim p_{\theta_2}(\cdot \mid v_2), z_3 \sim p_{\theta_3}(\cdot \mid v_3)$. Although each MI term is maximized by InfoNCE objective which is also adopted by our method, CMC differs from ours in the following aspects:

1. Each MI term $I_{\theta_i \theta_j}(Z_i; Z_j)$ in equation 11 encourages its input single-view representations to capture the shared factors of variation but not the view-specific factors. This is because $I_{\theta_i \theta_j}(Z_i; Z_j)$ is upper-bounded by $I(V_i; V_j)$ which quantifies the amount of **shared information**. On the other hand, each MI term $I_{\theta_s}(V_s; Z_s)$ in our objective (equation 1) encourages the subset-view representation $z_s$ to capture both shared and view-specific factors of variation in the subset of views $v_s$, which results in its optimal solution.

2. CMC lacks any mechanism to aggregate any subset of views other than naive approaches (e.g. concatenating or averaging single-view representations). In contrast, our method explicitly learns to aggregate any subset of views via IVW average of single-view representations based on their precision.

3. Due to its pairwise optimization, the computation cost of CMC grows quadratically with the number of views ($O(N^2)$), while our method scales linearly with the number of views ($O(N)$).

## B.2    GMC (POKLUKAR ET AL., 2022)

To learn the single-view representations and the complete-view representation at the same time, GMC optimizes the following contrastive objective function.

$$\sum_{n=1}^{N} \sum_{i=1}^{K} - \log \frac{e^{\frac{\langle z_n^{(i)}, z_{1:N}^{(i)} \rangle}{\tau}}}{\sum_j e^{\frac{\langle z_n^{(i)}, z_{1:N}^{(j)} \rangle}{\tau}} + \sum_{i \neq j} e^{\frac{\langle z_n^{(i)}, z_n^{(j)} \rangle}{\tau}}} - \log \frac{e^{\frac{\langle z_{1:N}^{(i)}, z_n^{(i)} \rangle}{\tau}}}{\sum_j e^{\frac{\langle z_{1:N}^{(i)}, z_n^{(j)} \rangle}{\tau}} + \sum_{i \neq j} e^{\frac{\langle z_{1:N}^{(i)}, z_{1:N}^{(j)} \rangle}{\tau}}},$$

$$(12)$$

where $\langle \cdot, \cdot \rangle$ denotes the inner product of its two input vectors. Although GMC aligns single-view representations $\{z_i\}_{i=1}^{N}$ with the complete-view representation $z_{1:N}$ at the computation cost that grows linearly with the number of views ($O(N)$), it differs from our method in the following aspects:

1. Similar to CMC, GMC also lacks any mechanism to aggregate any subset of views except the complete views. On the contrary, our method explicitly learns to aggregate any subset of views via IVW average.

2. GMC uses Cosine similarity (inner product in equation 12) which induces strong alignment of representations and thus discards view-specific information, while Mahalanobis distance is used in our method which helps capturing view-specific information as we observed in Section 4.1.

3. GMC requires twice more neural network parameters compared to ours, since $z_{1:N}$ of GMC is encoded by separate encoders which are independent of the encoders of $\{z_i\}_{i=1}^N$.

### B.3 MULTI-VIEW VAEs AND MoPoE-VAE (SUTTER ET AL., 2021)

Multi-View VAEs including MVAE (Wu & Goodman, 2018), MMVAE (Shi et al., 2019), and MoPoE-VAE (Sutter et al., 2021) commonly optimizes ELBO of the multi-view data below.

$$\mathbb{E}_{p_D(v_{1:N})} \left[ \sum_{i=1}^{N} \left[ \mathbb{E}_{p_\theta(z|v_{1:N})} \left[ \ln q_\phi^i (v_i|z) \right] \right] - D_{KL} \left[ p_\theta(z \mid v_{1:N}) \| N(0, \mathrm{I}) \right] \right], \tag{13}$$

where $q_\phi^i (v_i|z)$ is a decoder dedicated to $i$-th view.

While MVAE learns to aggregate complete views with its PoE (IVW) encoder, it struggles with calibrating each single-view encoder. In contrast, MMVAE explicitly optimizes each single-view encoder with its MoE joint encoder, although it fails to aggregate information from multiple views.

To overcome disadvantages of MVAE and MMVAE, MoPoE-VAE introduces the MoPoE joint encoder which combines MoE and PoE (further details regarding the structures of MoE, PoE, and MoPoE can be found in Section 3.1)). While MoPoE-VAE learns to aggregate any subset of views, which is similar to ours, it differs from our method in following aspects:

1. MoPoE-VAE implicitly calibrates all subset-view representations $z_s$ in the raw *view space* by making them infer the complete views. On the other hand, our method explicitly calibrates all $z_s$ in the *representation space* by making them infer the complete-view representation.

2. Optimization of equation 13 requires Multi-View VAEs to compute the density of its joint encoder, resulting in the computation cost of MoPoE-VAE that exponentially scales with the number of views $O(2^N)$. However, the optimization of our objective (equation 6) does not require the density computation of the joint encoder, resulting in the computation cost of ours that linearly scales with the number of views $O(N)$.

3. MoPoE-VAE relies on training decoders, while our method can be trained without decoders.

### B.4 MVTCAE (HWANG ET AL., 2021)

Borrowing the same encoder and decoder structures of MVAE, MVTCAE explicitly calibrates each single-view encoder in the representation space using its PoE joint encoder. Specifically, it optimizes the convex combination of the ELBO (equation 13) and the following objective.

$$\mathbb{E}_{p_D(v_{1:N})} \left[ \frac{1}{N} \sum_{i=1}^{N} \left[ (N-1) \mathbb{E}_{p_{\theta_{1:N}}(z_{1:N}|v_{1:N})} \left[ \ln q_\phi^v (v_i|z_{1:N}) \right] - D_{KL} \left[ p_{\theta_{1:N}}(z_{1:N}|v_{1:N}) \| p_{\theta_i}(z_i|v_i) \right] \right] \right]. \tag{14}$$

Although MVTCAE is trainable only at the computation cost that scales linearly with the number of views $O(N)$, it can be easily distinguished from ours by the following aspects:

1. MVTCAE does not learn any subset-view representations except single-view and complete-view representations during training, while our method explicitly learns all subset-view representations.

2. MVTCAE relies on training decoders, while our method can be trained without decoders.

## C   COMPUTATION RESOURCES

10 systems equipped with following devices were used in all our experiments.

CPU: Intel(R) Core(TM) i7-9700K CPU @ 3.60GHz

Memory: 32 Gb.

GPU: TITAN V

## D   DETAILS IN EXPERIMENTS

We report the details in our experiment including the hyperparameters and the network structures.

**Hyperparameter search**   In all 3 sets of experiments, we trained each method using Adam (Kingma & Ba, 2015) optimizer with learning rate $1e^{-4}$ and batch size 256, which ensures that all methods converge. For each method, we carefully searched for any method-specific hyperparameters as below.

For CMC (Tian et al., 2020), GMC (Poklukar et al., 2022), and GMCs, we searched for their optimal temperature $\tau$ in $\{0.01, 0.02, 0.05, 0.1, 0.2, 0.3, 0.5\}$ in each dataset, which completely includes all the values suggested by GMC (Poklukar et al., 2022) and densely covers the range suggested by CMC (Tian et al., 2020).

For MoPoE-VAE (Sutter et al., 2021) and MVTCAE (Hwang et al., 2021), we searched for their optimal $\beta$, the coefficient of their KL terms, in $\{0.1, 0.3, 0.5, 0.7, 1.0, 3.0, 5.0\}$ in synthetic dataset and 4 datasets from MultiBench (Liang et al., 2021). In Caltech-101, we applied the optimal hyperparameter settings found by Hwang et al. (2021).

For CwA and CwA+recon, we searched for the optimal temperature $\tau$ in $\{4.0, 8.0, 12.0, 16.0, 20.0\}$ and $\beta$ in $\{0.1, 0.3, 0.5, 0.7, 1.0, 3.0, 5.0\}$ in each dataset.

### D.1   SYNTHETIC DATASET

**Dataset**   We generated a synthetic dataset composed of 10,000 instances of 8 views. For each instance, 2 types of data-generative factors are sampled: a view-specific factor $g_i \sim [0, 2]$ for each view ($1 \leq i \leq 8$) and a shared factor $g_s \sim [-1, 1]$. Each view $v_i \in \mathbb{R}^{100}$ was generated by drawing 100 samples from a Gaussian distribution $N(g_s, g_i^2)$, resulting in vectorized views. The dataset was split into train(8,000), valid(1,000), test(1,000) sets where each of values in the parentheses represents the number of samples. The data generation process is described in Figure 5.

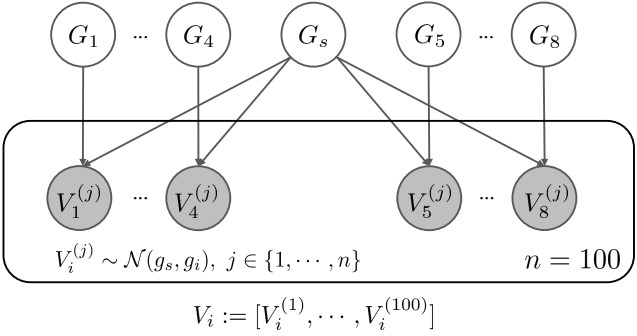

$$V_i := [V_i^{(1)}, \cdots, V_i^{(100)}]$$

Figure 5: Data generation process for the synthetic dataset.

**Remark**   The purpose of using the synthetic dataset is two-fold: (1) to ensure the presence of both a shared factor observable across all views and view-specific factors unique to each view, and (2) to evaluate whether each MVRL method can effectively capture both types of information. Regression

on those 9 factors as a downstream task provides a controlled way to evaluate whether a given MVRL method captures all important information in its representations. While this dataset may lack direct practical applicability, we believe that its ability to offer a controlled setup for verifying key capabilities (e.g., aggregating complementary information across views) makes it valuable for tasks with real-world analogs, such as autonomous driving, medical diagnostics, or multi-sensor systems.

**Implementation detail**    Every $v_i$ was encoded by its dedicated MLP encoder into a single-view representation $z_i \in \mathbb{R}^9$. We set the size of the representation of each method to be 9, considering that there are 9 true data generative factors (8 view-specific factors + 1 shared factors). The network structures and sizes of each method can be found in Table 4 and Table 5. Enc, Dec, Det, and Proj stand for encoder, decoder, deterministic, and projection respectively. Further details can be found in the submitted code (please find the directory named Syn).

| Method | CMC | CwA (Ours) | MoPoE-VAE, MVTCAE |
|---|---|---|---|
| | | | |
| Network | Det.Enc. $\theta_n$ | Enc. $p_{\theta_n}(z\|v_n)$ | Enc. $p_{\theta_n}(z\|v_n)$ |
| Input | $v_n$ | $v_n$ | $v_n$ |
| Layer 1 | FC. 256. ReLU | FC. 256. ReLU | FC. 256. ReLU |
| Layer 2 | FC. 256. ReLU | FC. 256. ReLU | FC. 256. ReLU |
| Layer 3 | FC. 9 | $2\times$ FC. 9 | $2\times$ FC. 9 |
| Output | $z_n$ | $\mu_n, \log \sigma_n^2$ | $\mu_n, \log \sigma_n^2$ |
| | | | |
| Network | | | Dec. $q_\phi(v_n\|z)$ |
| Input | | | $z \sim p_\theta(z\|v_{1:8})$ |
| Layer 1 | | | FC. 256. ReLU |
| Layer 2 | | | FC. 256. ReLU |
| Layer 3 | | | FC. 100. |
| Output | | | $\hat{v}_n$ |

Table 4: Network structures of CMC, CwA(Ours), MoPoE-VAE, and MVTCAE in synthetic dataset.

| Method | GMCs | | GMC | |
|---|---|---|---|---|
| | | | | |
| Network | Det.Enc. $\theta_n$ | | Det.Enc. $\theta_n^{(1)}$ | Det.Enc. $\theta_n^{(2)}$ |
| Input | $v_n$ | | $v_n$ | $v_n$ |
| Layer 1 | FC. 256. ReLU | | FC. 256. ReLU | FC. 256. ReLU |
| Layer 2 | FC. 256. ReLU | | FC. 256. ReLU | FC. 256. ReLU |
| Output | $h_n$ | | $h_n^{(1)}$ | $h_n^{(2)}$ |
| | | | | |
| Network | Linear projection $\psi$ | Shared Enc. $\phi$ | Linear Proj. $\psi$ | Shared Enc. $\phi$ |
| Input | $[h_1, ..., h_8]$ | $h_1,...,h_8$, or $h_{1:8}$ | $[h_1^{(1)}, ..., h_8^{(1)}]$ | $h_1^{(2)},...,h_8^{(2)}$, or $h_{1:8}^{(1)}$ |
| Layer 1 | FC. 256 | FC. 9 | FC. 256 | FC. 9 |
| Output | $h_{1:8}$ | $z_1,...,z_8$, or $z_{1:8}$ | $h_{1:8}^{(1)}$ | $z_1,...,z_8$, or $z_{1:8}$ |

Table 5: Network structures of GMC and GMCs in synthetic dataset.

### D.2 MULTIBENCH

**Datasets** MOSI (Zadeh et al., 2016), MUSTARD (Castro et al., 2019), FUNNY (Hasan et al., 2019), and MOSEI (Zadeh et al., 2018) are realistic datasets commonly composed of 3 views, video(V), audio(A), and text(T). These datasets are preprocessed and released in public by Multi-Bench (Liang et al., 2021). We summarize the statistics of the datasets below.

1. MOSI
   2,199 samples of the triplets (V: 45×20 dim., A: 45×5 dim., T: 45×300 dim.).

2. MUSTARD
   690 samples of the triplets (V: 50×371 dim., A: 50×81 dim., T: 50×300 dim.).

3. FUNNY
   16,514 samples of the triplets (V: 20×371 dim., A: 20×81 dim., T: 20×300 dim.).

4. MOSEI
   22,777 samples of the triplets (V: 50×35 dim, A: 50×74 dim., T: 50×300 dim.).

**Implementation detail** For all methods, we used Transformer encoder for each view with hidden size 200 and representation size 128. Following the implementation of MVAE (Wu & Goodman, 2018) provided by MultiBench (Liang et al., 2021), we used Timeseries decoder for each view in MoPoE-VAE and MVTCAE. Detailed information on the size and structure of networks is summarized in Table 6 and Table 7. Enc, Dec, Det, Proj, TSDec. stand for encoder, decoder, deterministic, projection, and Timeseries decoder respectively. Further details can be found in the submitted code (please see the directory named Multi).

| Method | CMC | CwA (Ours) | MoPoE-VAE, MVTCAE |
|---|---|---|---|
| | | | |
| Network | Det.Enc. $\theta_n$ | Enc. $p_{\theta_n}(z\|v_n)$ | Enc. $p_{\theta_n}(z\|v_n)$ |
| Input | $v_n$ | $v_n$ | $v_n$ |
| Layer 1 | Transformer. 200 | Transformer. 200 | Transformer. 200 |
| Layer 2 | FC. 128 | 2× FC. 128 | 2× FC. 128 |
| Output | $z_n$ | $\mu_n, \log\sigma_n^2$ | $\mu_n, \log\sigma_n^2$ |
| | | | |
| Network | | | Dec. $q_\phi(v_n\|z)$ |
| Input | | | $z \sim p_\theta(z\|v_{1:3})$ |
| Layer 1 | | | TSDecoder, dim.($v_n$) |
| Layer 2 | | | FC. dim.($v_n$) |
| Output | | | $\hat{v}_n$ |

Table 6: Network structures of CMC, CwA, MoPoE-VAE, and MVTCAE in 4 MultiBench datasets.

| Method | GMCs | | GMC | |
|---|---|---|---|---|
| | | | | |
| Network | Det.Enc. $\theta_n$ | | Det.Enc. $\theta_n^{(1)}$ | Det.Enc. $\theta_n^{(2)}$ |
| Input | $v_n$ | | $v_n$ | $v_n$ |
| Layer 1 | Transformer. 200 | | Transformer. 200 | Transformer. 200 |
| Output | $h_n$ | | $h_n^{(1)}$ | $h_n^{(2)}$ |
| | | | | |
| Network | Linear projection $\psi$ | Shared Enc. $\phi$ | Linear Proj. $\psi$ | Shared Enc. $\phi$ |
| Input | $[h_1;h_2;h_3]$ | $h_1, h_2, h_3,$ or $h_{123}$ | $[h_1^{(1)};h_2^{(1)};h_3^{(1)}]$ | $h_1^{(2)}, h_2^{(2)}, h_3^{(2)},$ or $h_{123}^{(1)}$ |
| Layer 1 | FC. 200 | FC. 128 | FC. 200 | FC. 128 |
| Output | $h_{123}$ | $z_1, z_2, z_3,$ or $z_{123}$ | $h_{123}^{(1)}$ | $z_1, z_2, z_3,$ or $z_{123}$ |

Table 7: Network structures of GMC and GMCs in 4 MultiBench datasets.

**Standard error of the performance in MultiBench** :

| Dataset | Model | 1 view | | | | 2 views | | | | 3 views |
|---|---|---|---|---|---|---|---|---|---|---|
| | | Video | Audio | Text | Avg. | V,A | V,T | A,T | Avg. | V,A,T |
| MOSI | CMC | ±0.55 | ±0.48 | ±0.4 | ±0.21 | ±0.68 | ±0.56 | ±0.31 | ±0.31 | ±0.3 |
| | GMC | ±0.98 | ±0.45 | ±1.05 | ±0.62 | ±0.72 | ±1.14 | ±0.91 | ±0.83 | ±0.64 |
| | GMCs | ±0.77 | ±0.53 | ±0.62 | ±0.23 | ±0.32 | ±0.66 | ±0.63 | ±0.45 | ±0.8 |
| | MoPoE | ±0.62 | ±0.58 | ±0.81 | ±0.38 | ±0.61 | ±0.86 | ±0.39 | ±0.42 | ±0.49 |
| | MVTCAE | ±0.6 | ±0.84 | ±0.82 | ±0.42 | ±0.51 | ±0.65 | ±0.9 | ±0.57 | ±0.7 |
| | CwA (Ours) | ±0.6 | ±0.46 | ±0.52 | ±0.19 | ±0.59 | ±0.48 | ±0.66 | ±0.31 | ±0.47 |
| MUSTARD | CMC | ±0.7 | ±0.9 | ±1.44 | ±0.57 | ±0.7 | ±1.33 | ±1.1 | ±0.78 | ±0.79 |
| | GMC | ±0.53 | ±1.16 | ±0.64 | ±0.34 | ±0.65 | ±0.83 | ±0.66 | ±0.53 | ±0.76 |
| | GMCs | ±1.63 | ±0.88 | ±0.67 | ±0.8 | ±1.21 | ±0.95 | ±0.58 | ±0.82 | ±0.74 |
| | MoPoE | ±2.0 | ±1.93 | ±2.11 | ±0.98 | ±1.01 | ±2.18 | ±2.13 | ±1.0 | ±0.58 |
| | MVTCAE | ±2.12 | ±2.22 | ±1.52 | ±1.14 | ±2.14 | ±2.02 | ±2.23 | ±0.93 | ±1.27 |
| | CwA (Ours) | ±1.17 | ±0.88 | ±0.62 | ±0.64 | ±0.78 | ±0.73 | ±0.53 | ±0.54 | ±0.88 |
| FUNNY | CMC | ±0.43 | ±0.38 | ±0.53 | ±0.24 | ±0.19 | ±0.25 | ±0.27 | ±0.12 | ±0.39 |
| | GMC | ±0.41 | ±0.53 | ±0.26 | ±0.23 | ±0.34 | ±0.25 | ±0.36 | ±0.27 | ±0.29 |
| | GMCs | ±0.38 | ±0.38 | ±0.47 | ±0.27 | ±0.3 | ±0.41 | ±0.39 | ±0.3 | ±0.5 |
| | MoPoE | ±0.8 | ±0.66 | ±0.33 | ±0.22 | ±0.31 | ±0.5 | ±0.28 | ±0.22 | ±0.32 |
| | MVTCAE | ±0.38 | ±0.56 | ±0.28 | ±0.24 | ±0.24 | ±0.27 | ±0.22 | ±0.13 | ±0.18 |
| | CwA (Ours) | ±0.36 | ±0.83 | ±0.38 | ±0.32 | ±0.45 | ±0.37 | ±0.24 | ±0.24 | ±0.26 |
| MOSEI | CMC | ±0.5 | ±0.2 | ±0.34 | ±0.15 | ±0.12 | ±0.26 | ±0.26 | ±0.16 | ±0.16 |
| | GMC | ±0.17 | ±0.14 | ±0.13 | ±0.09 | ±0.06 | ±0.08 | ±0.08 | ±0.05 | ±0.12 |
| | GMCs | ±0.24 | ±0.17 | ±0.15 | ±0.09 | ±0.1 | ±0.12 | ±0.13 | ±0.06 | ±0.16 |
| | MoPoE | ±5.26 | ±5.56 | ±5.92 | ±2.99 | ±5.09 | ±4.7 | ±0.55 | ±1.79 | ±0.16 |
| | MVTCAE | ±4.72 | ±4.48 | ±3.85 | ±2.53 | ±4.68 | ±1.26 | ±5.33 | ±2.33 | ±0.12 |
| | CwA (Ours) | ±0.67 | ±0.15 | ±0.5 | ±0.15 | ±0.09 | ±0.25 | ±0.2 | ±0.14 | ±0.19 |

Table 8: Standard error (%) of the learned representation from subset views in 4 MultiBench datasets.

### D.3 CALTECH-101

**Dataset**  Caltech-101 (Fei-Fei et al., 2004) is a collection of 101 classes of images designed for learning object recognition tasks. Li et al. (2015) extracted 6 visual features from 9,144 images in Caltech-101 to compile a multi-view dataset and released them in public. Those 6 visual features are listed along with their dimensions below.

1. Gabor feature (Oliva & Torralba, 2001): 48 dimensions.

2. Wavelet moments (Oliva & Torralba, 2001): 40 dimensions.

3. CENTRIST (Wu & Rehg, 2010): 254 dimensions.

4. Histogram of Oriented Gradients (Dalal & Triggs, 2005): 1,984 dimensions.

5. GIST (Oliva & Torralba, 2001): 512 dimensions.

6. Local Binary Pattern (Ojala et al., 2002): 928 dimensions.

Following MVTCAE (Hwang et al., 2021), we standardized each feature using Scikit learn (Pedregosa et al., 2011).

**Implementation detail**  We used the network architecture same as MVTCAE except the size of the output feature $h$ of FC layer and the size of the representation. We increased the feature size from 200 to 512 and the representation size from 100 to 256, since this setting was commonly beneficial for all comparing methods. Detailed information on the size and structure of networks is summarized in Table 9 and Table 10. Further details can be found in the submitted code (please find the directory named Cal).

| Method | CMC | CwA (Ours) | MoPoE-VAE, MVTCAE, CwA+recon (Ours) |
|---|---|---|---|
| | | | |
| Network | Det.Enc. $\theta_n$ | Enc. $p_{\theta_n}(z\|v_n)$ | Enc. $p_{\theta_n}(z\|v_n)$ |
| Input | $v_n$ | $v_n$ | $v_n$ |
| Layer 1 | FC. 512. ReLU | FC. 512. ReLU | FC. 512. ReLU |
| Layer 2 | FC. 256 | $2\times$ FC. 256 | $2\times$ FC. 256 |
| Output | $z_n$ | $\mu_n, \log \sigma_n^2$ | $\mu_n, \log \sigma_n^2$ |
| | | | |
| Network | | | Dec. $q_\phi(v_n\|z)$ |
| Input | | | $z \sim p_\theta(z\|v_{1:6})$ |
| Layer 1 | | | FC. 512. ReLU |
| Layer 2 | | | FC. dim.$(v_n)$ |
| Output | | | $\hat{v}_n$ |

Table 9: Network structures of CMC, CwA(Ours), MoPoE-VAE, and MVTCAE in Caltech-101 dataset.

| Method | GMCs | | GMC | |
|---|---|---|---|---|
| | | | | |
| Network | Det.Enc. $\theta_n$ | | Det.Enc. $\theta_n^{(1)}$ | Det.Enc. $\theta_n^{(2)}$ |
| Input | $v_n$ | | $v_n$ | $v_n$ |
| Layer 1 | FC. 512. ReLU | | FC. 512. ReLU | FC. 512. ReLU |
| Output | $h_n$ | | $h_n^{(1)}$ | $h_n^{(2)}$ |
| | | | | |
| Network | Linear projection $\psi$ | Shared Enc. $\phi$ | Linear Proj. $\psi$ | Shared Enc. $\phi$ |
| Input | $[h_1; ...; h_6]$ | $h_1,...,h_6$, or $h_{1:6}$ | $[h_1^{(1)}; ...; h_6^{(1)}]$ | $h_1^{(2)},...,h_6^{(2)}$, or $h_{1:6}^{(1)}$ |
| Layer 1 | FC. 512 | FC. 256 | FC. 512 | FC. 256 |
| Output | $h_{1:6}$ | $z_1,...,z_6$, or $z_{1:6}$ | $h_{1:6}^{(1)}$ | $z_1,...,z_6$, or $z_{1:6}$ |

Table 10: Network structures of GMC and GMCs in Caltech-101 dataset.

# E  ADDITIONAL EXPERIMENT RESULTS

## E.1  HYPERPARAMETER SENSITIVITY

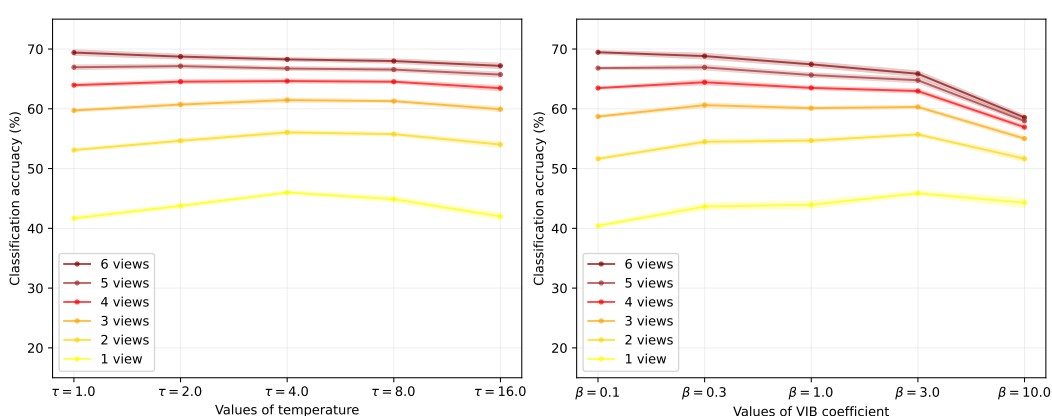

Figure 6: Sensitivity of CwA to hyperparameters $\tau$ (left) and $\beta$ (right).

To investigate the sensitivity of our method to hyperparameters, we report the performance of CwA on the Caltech-101 dataset with varying $\beta$ (the coefficient of the Variational Information Bottleneck (VIB)) and $\tau$ (the temperature of the InfoNCE objective) in Figure 6. Specifically, we varied $\tau$ from 1 to 16 with fixed $\beta = 1.0$ (left) and varied $\beta$ from 0.1 to 10.0 with fixed $\tau = 4.0$ (right). We observe that CwA is not sensitive to the choice of $\tau$, while increasing $\beta$ to high values, such as 10, can be critical. This is expected as $\beta$ controls the strength of the VIB, which penalizes the amount of information encoded in each single-view representation. Consequently, each single-view encoder is forced to discard important information when excessively high values of are used.

## E.2  VISUALIZATION OF THE LEARNED REPRESENTATIONS

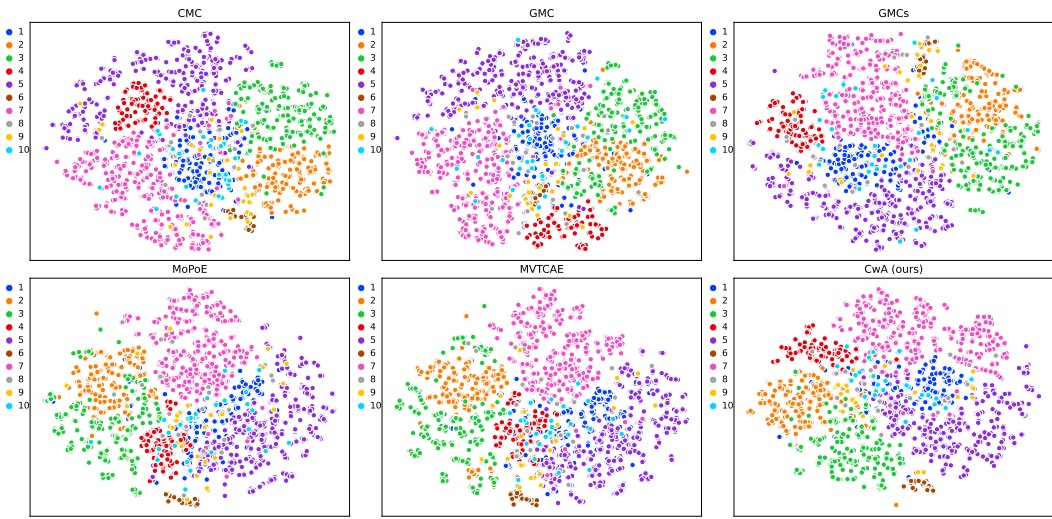

Figure 7: T-SNE visualization of the representations. From the top left corner to the bottom right corner, results are from CMC, GMC, GMCs, MoPoE, MVTCAE, and CwA.

Figure 7 shows the t-SNE feature visualization of our method and compared them with other methods on Caltech-101. Among the 101 classes in the Caltech101 dataset, we collected samples from the first 10 classes and extracted representations using each model. Figure 7 summarizes the results. We observed that all methods struggle to separate samples from classes 8 (gold), 9 (grey), and 10

(sky blue). Additionally, MVTCAE, MoPoE, GMC, and GMCs commonly fail to differentiate samples from classes 2 (orange) and 3 (green). Similarly, CMC fails to separate samples from classes 4 (red) and 7 (pink). On the other hand, our method successfully separates samples from classes 2, 3, 4, and 7, demonstrating its effectiveness.

### E.3 TRAINING FROM MISSING VIEWS

Although our primary focus is learning representations from complete-view training data (MVRL), our method can also learn from training data with missing views (Partial MVRL (Hwang et al., 2021)). This is due to its encoder structure, which can encode any available views via IVW average, similar to MVTCAE and MoPoE-VAE. We investigated the effectiveness of this simple idea for Partial MVRL scenario on Caltech-101, using the same evaluation protocol in Section 4.3 but dropping each view in training and validation on Caltech-101 data with probability 0.5.

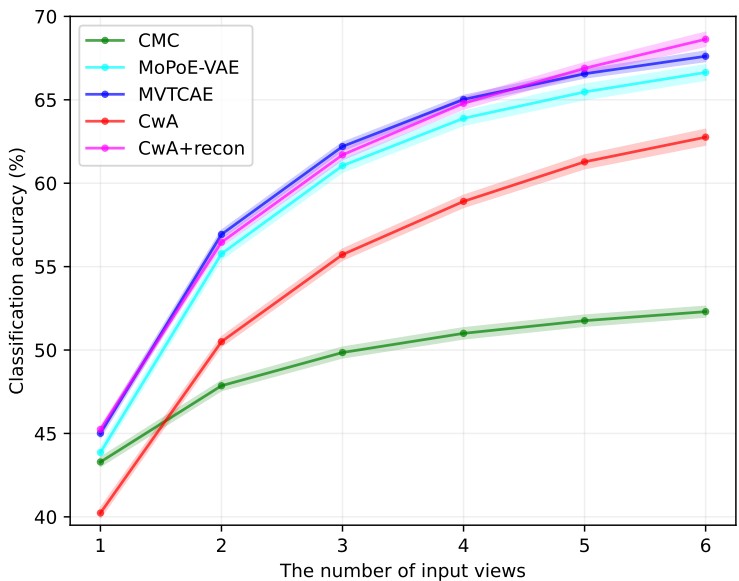

Figure 8: Results of training with missing views (Partial MVRL) on Caltech-101.

Figure 8 summarizes the result. GMC and GMCs are not included in the set of baseline methods since their complete-view representations strictly require the presence of all views. We observed that CwA significantly outperforms CMC when at least two views are given, demonstrating its effectiveness to Partial MVRL. Furthermore, when combined with Auto-Encoders, our method (CwA+recon) shows strong results, performing similarly to MVTCAE and outperforms MoPoE-VAE.

We intend to further improve CwA for Partial MVRL, which we leave as future work.

## E.4 Comparison to multi-view clustering methods

We additionally evaluated DCP (Lin et al., 2022) and InfoDDC (Trosten et al., 2023), which are mutli-view clustering methods that support more than two views. Specifically, DCP can learn single-view representations without labels by applying the three components to every pair of views: (1) within-view reconstruction, (2) cross-view contrastive learning, and (3) cross-view latent prediction. InfoDDC also learns single-view representations through cross-view contrastive learning between every pair of views. Furthermore, it learns the weight of each single-view representation using its unsupervised clustering objective, resulting in the weighted average of single-view representations as its complete-view representation.

Based on the official implementations of DCP and InfoDDC, we carefully tuned their hyperparameters, including coefficients of loss terms, temperature values related to their contrastive learning, and the number of clusters (to be equal the number of labels).

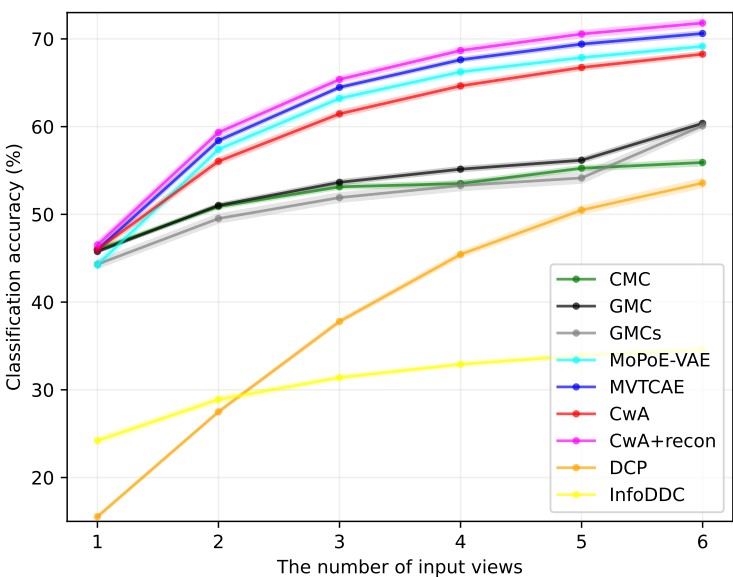

Figure 9: Results on Caltech-101 with additional baseline methods (DCP, InfoDDC)

Figure 9 summarizes the evaluation of DCP and InfoDDC on the Caltech-101 dataset. DCP shows poor performance when only a few views are given because it relies on predicting the representation of missing views from that of available views. Considering that the amount of information in each view varies significantly in Caltech-101, it might be difficult to reconstruct the representations of HOG (1984 dim), GIST (512 dim), or LBP (928 dim) views given the representations of Gabor (48 dim) and WM (40 dim) views. On the other hand, InfoDDC fails to show competitive performance with any number of views. We hypothesize that this is because (1) learning clusters without labels in their formulation is complicated when there are many classes of labels and (2) the contrastive loss between every pair of views tends to discard view-specific information.

While DCP and InfoDDC show different trends, they significantly underperform in common compared to other methods.

1404
1405
1406
1407
1408
1409
1410
1411
1412
1413
1414
1415
1416
1417
1418
1419
1420
1421
1422
1423
1424
1425
1426
1427
1428
1429
1430
1431
1432
1433
1434
1435
1436
1437
1438
1439
1440
1441
1442
1443
1444
1445
1446
1447
1448
1449
1450
1451
1452
1453
1454
1455
1456
1457

### E.5 EVALUATION ON HANDWRITTEN

We conducted an additional evaluation on the Handwritten (Li et al., 2015) dataset with six hand-crafted feature views, as it has the most views among the suggested datasets. In addition to MVT-CAE, MoPoE-VAE, GMC, GMCs, and CMC, we included additional baseline methods such as DCP and InfoDDC.

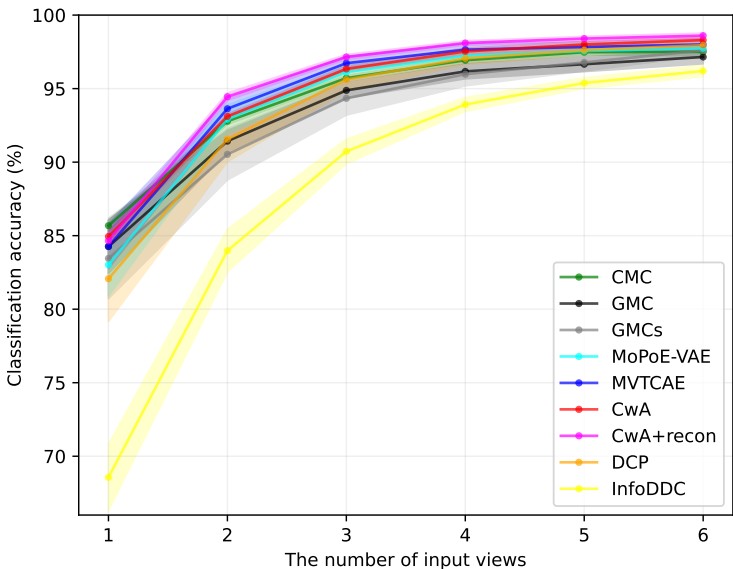

Figure 10: Results on Handwritten with additional baseline methods (DCP, InfoDDC)

Figure 10 summarizes the results. Unlike the results on Caltech-101, we observed that most of the baseline methods show competitive performance. This is because the Handwritten dataset has much lower-dimensional views (at most 240 in Handwritten versus 1984 in Caltech-101) and fewer classes (10 in Handwritten versus 101 in Caltech-101).

Although the performance gain is less significant compared to the results on Caltech-101, CwA+recon still shows the best performance among all compared methods when at least two views are given. Similarly, CwA outperforms all contrastive learning methods that lack decoders such as CMC, GMC, GMCs, and InfoDDC, given at least two views.

## E.6 EVALUATION ON IMAGENET-100

We conducted additional experiments on the ImageNet-100 dataset, which consists of 100 subsampled classes selected by CMC from the full ImageNet dataset. Following the experimental setup in CMC, we transformed RGB images into two views in different color spaces (L and ab).

To accommodate our limited computational resources, we used pretrained ResNet34 [4] to extract 512-dimensional deep features from each view. We then trained each representation learning method with MLP encoders to learn 128-dimensional representations from these features. For evaluation, we followed the same linear classification protocol as in Sections 4.2 and 4.3, training a linear classifier to predict image labels from the frozen representations.

Table 11 below summarizes the results. The results demonstrate that all methods, including ours, benefit from the addition of views. Notably, our method achieves superior performance compared to all baseline methods for both single-view inputs (L, ab) and two-view inputs (Lab). These findings indicate that our method can effectively learn representations from realistic datasets, providing further validation of its generalizability and effectiveness.

| Method | L | ab | Lab |
|---|---|---|---|
| CMC | 81.50±0.05 | 71.12±0.04 | 83.8±0.04 |
| GMC | 81.59±0.07 | 71.05±0.11 | 83.20±0.13 |
| GMCs | 80.10±0.05 | 69.62±0.13 | 82.55±0.09 |
| MoPoE | 80.30±0.04 | 68.69±0.09 | 81.66±0.14 |
| MVTCAE | 81.32±0.12 | 70.12±0.18 | 83.13±0.17 |
| CwA | **82.1**±0.07 | **71.75**±0.06 | **84.75**±0.06 |

Table 11: Classification accuracy (%) of the learned representation from all combination of views on ImageNet-100.

## E.7 MSE FOR JOINT PREDICTION AND ABLATION STUDY

(a) The performance of comparing methods.

(b) Ablation study.

Figure 11: Results of linear regression in Synthetic dataset. Mean squared error between true data-generative factors and predicted factors is measured with incrementally adding views.

**Evaluation protocol**   We trained linear regression models to predict all data-generative factors using the frozen representation. Specifically, we pretrained each method using the train set for 1,000 epochs, validating every 10 epochs. During validation, we trained a linear regression model with z of the complete views to predict true data-generative factors $[g_1; ...; g_8; g_s]$ in the train set. We then evaluated it using z of the complete views in the validation set. We saved the regression model and each method when their performance was the best in the validation set. After training, we evaluated the saved models by measuring the mean squared error between true generative factors and those predicted from z of accumulated input views (e.g. view 1, views 1+2, ..., views 1:N) in the test set.

**Result**   Figure 11a compares the performance of all methods. The x-axis represents the input view(s) accumulating one by one, and the y-axis represents the mean squared error. While CwA slightly underperforms with a single view, it effectivley improves its performance when multiple views are available. As a result, CwA significantly outperforms all baseline methods when more than 2 views are given. The result demonstrates that our method better captures all factors of variation. Conversely, the other methods commonly fail to leverage additional views. CL methods are encouraged by their objective function to discard view-specific information, so additional views help only in identifying the shared factors. Furthermore, due to the reconstruction of input views, VAE methods capture noise incurred by sampling views in the data generation process rather than discovering true underlying factors, leading to poor performance in downstream tasks.

**Ablation study**   To assess the impact of the joint encoder choice in CwA, we conducted an ablation study with different encoder choices. Specifically, we trained MoE and PoE joint encoder by optimizing our objective equation 6. Figure 11b shows the results. While the MoE joint encoder shows competitive performance with a single view, its performance barely improves with the additional views. This is because MoE combines only the single-view encoders but not the rest of the subset-view encoders, thus maximizing MIs only between single views and their representations as discussed in Section 3.2. Consequently, its ability to extract representations from multiple views is limited. In contrast, the PoE joint encoder exhibits monotonic improvement by leveraging additional views, though its performance is still limited compared to ours. This is because it learns to extract information from all views by maximizing MI between the complete views and the complete-view representation, but does not consider extracting from other subsets of views as discussed in Section 3.2. The results indicate that the MoPoE joint encoder, which optimizes all combinations of views by maximizing MI between every subset of views and its representations, is advantageous.

# F PSEUDOCODE

---

**Algorithm 1** CwA. Each step is specified with its computation cost w.r.t. the number of views.

---

1: **Input:** $K$: batch size, $N$: # of views, $D$: dataset $\{v_{1:N}^{(i)}\}_{i=1}^{|D|}$, $\{\text{Enc}_{\theta_i}\}_{i=1}^N$: encoders for $N$ views.
2: **for** sampled minibatch $\{v_{1:N}^{(k)}\}_{k=1}^K \sim D$ **do**
3:     **for** $k = 1$ **to** $K$ **do**
4:        $\{\mu_{\theta_i}^{(k)}, \sigma_{\theta_i}^{(k)}\}_{i=1}^N = \{\text{Enc}_{\theta_i}(v_i^{(k)})\}_{i=1}^N$           // encode each view $v_i$ ($O(N)$)
5:        $\mu_{1:N}^{(k)}, \sigma_{1:N}^{(k)} = \text{IVW}(\{\mu_{\theta_i}^{(k)}, \sigma_{\theta_i}^{(k)}\}_{i=1}^N)$     // enc. $v_{1:N}$ via IVW of all 1-view representations ($O(N)$)
6:        $\mu_s^{(k)}, \sigma_s^{(k)} = \text{IVW}(\text{uniform\_subsample}(\{\mu_{\theta_i}^{(k)}, \sigma_{\theta_i}^{(k)}\}_{i=1}^N))$ // enc. random subset $v_s$ via IVW ($O(N)$)
7:        $z^{(k)} \sim N(\mu_s^{(k)}, (\sigma_s^{(k)})^2 \cdot \mathbf{I})$             // sample $z$ from $p_{\theta_s}(z_s|v_s)$ ($O(1)$)
8:     **end for**
9:     **define** $f(z, \mu, \sigma) := -\frac{(z-\mu)^T \sigma^{-2} \mathbf{I}(z-\mu)}{\tau}$       // define $f$ to measure Mahalanobis distance
10:     **for** $k = 1$ **to** $K$ **do**
11:        $L_{VIB}^{(k)} = \sum_{i=1}^N D_{KL}[N(\mu_{\theta_i}^{(k)}, (\sigma_{\theta_i}^{(k)})^2 \cdot \mathbf{I}) || N(0, \mathbf{I})]$      // compute VIB loss of each view ($O(N)$)
12:        $L_{Contrast}^{(k)} = -\log \frac{e^{f\left(z^{(k)}, \mu_{1:N}^{(k)}, \sigma_{1:N}^{(k)}\right)}}{\sum_{j=1}^K e^{f\left(z^{(k)}, \mu_{1:N}^{(j)}, \sigma_{1:N}^{(j)}\right)}}$        // compute InfoNCE loss ($O(1)$)
13:     **end for**
14:     Update $\{\theta_i\}_{i=1}^N$ to minimize $L = \frac{1}{K} \sum_{k=1}^K L_{Contrast}^{(k)} + \frac{\beta}{2^N - 1} L_{VIB}^{(k)}$
15: **end for**

---

# G DISCUSSION ON INVERSE-VARIANCE WEIGHTED AVERAGE

Inverse-Variance Weighted (IVW) average (Cochran & Carroll, 1953; Cochran, 1954; Hinton, 2002) is a classical (late) fusion method widely used in statistical sensor fusion. When observations are generated from the underlying state with independent Gaussian random noise, the IVW average serves as the Maximum Likelihood Estimation (MLE) of the state. To assess its optimality within the context of MVRL, let $\mu$ represent the ground-truth state that encapsulates all factors of variation for a given instance.

Assuming that the mean of each single-view representation $\mu_i$ from the i-th view is an observation of $\mu$ with Gaussian noise $\sigma_i^2$ s.t. $p(\mu_i|\mu) = N(\mu, \sigma_i^2 I)$, the IVW average $\mu_{1:N} = \frac{\sum_{i=1}^N \mu_i/\sigma_i^2}{\sum_{i=1}^N 1/\sigma_i^2}$ is the MLE of $\mu$, which we show below:

*Proof.* Since log of a probability function is monotonically increasing, the MLE of $\mu$ is obtained by maximizing the log-likelihood of $\{\mu_i\}_{i=1}^N$, given by:

$$\log p(\{\mu_i\}_{i=1}^N|\mu) = \log \left(\prod_{i=1}^N p(\mu_i|\mu)\right) = \sum_{i=1}^N \log p(\mu_i|\mu) = \sum_{i=1}^N \left(-\frac{1}{2}\log\left(2\pi\sigma_i^2\right) - \frac{(\mu_i - \mu)^2}{2\sigma_i^2}\right).$$

Since $\log p(\{\mu_i\}_{i=1}^N|\mu)$ is a concave function of $\mu$, it reaches its maximum when its derivative is zero:

$$\frac{d\log p(\{\mu_i\}_{i=1}^N|\mu)}{d\mu} = \frac{d}{d\mu}\sum_{i=1}^N \left(-\frac{1}{2}\log\left(2\pi\sigma_i^2\right) - \frac{(\mu_i - \mu)^2}{2\sigma_i^2}\right) = 0.$$

Since the first term inside of the summation is constant w.r.t. $\mu$, we have:

$$\sum_{i=1}^N \frac{\mu - \mu_i}{\sigma_i^2} = 0, \quad \sum_{i=1}^N \frac{\mu}{\sigma_i^2} = \sum_{i=1}^N \frac{\mu_i}{\sigma_i^2}.$$

Finally, rearranging gives:

$$\mu_{\text{MLE}} = \frac{\sum_{i=1}^{N} \frac{\mu_i}{\sigma_i^2}}{\sum_{i=1}^{N} \frac{1}{\sigma_i^2}} = \mu_{1:N}.$$

Thus, $\mu_{1:N}$, the IVW average of $\{\mu_i\}_{i=1}^{N}$ is the MLE of $\mu$. $\qquad\square$

The assumption s.t. $p(\mu_i|\mu) = N(\mu, \sigma_i^2 \mathrm{I})$ aligns naturally with our method, as all $\mu_i$ are derived from the same instance but exhibit different precisions based on the characteristics of each view. These precisions are captured by the learned variance $\sigma_i^2$ of each single-view representation. Consequently, IVW allows each single-view representation to contribute to the complete-view representation in proportion to its precision $\frac{1}{\sigma_i^2}$, making it an optimal choice for MVRL.

Beyond its statistical foundation, IVW offers computational scalability, with costs that scale linearly with the number of input views. This aligns with our goal of retaining scalability in MVRL. While alternatives like cross-view attention-based fusion could be considered, they incur a quadratic computational cost relative to the number of input views, making them less practical for scenarios involving many views. Moreover, such approaches have been empirically shown to perform less effectively when handling subset views, as observed in Hwang et al. (2023).

