# OpenReview forum: "Contrast with Aggregation: A Scalable Framework for Multi-View Representation Learning"
_ICLR.cc/2025/Conference — Submitted to ICLR 2025_

### Official Review · Reviewer_48cg · 2024-11-01

**Soundness:** 3
**Presentation:** 3
**Contribution:** 3
**Rating:** 8
**Confidence:** 2

**Summary:**

This paper studies an innovative multi-view representation learning framework：Contrast with Aggregation: A scalable Framework for Multi-view Representation Learning. Existing multi-view representation learning methods have three main limitations: (1) discarding potentially valuable view-specific information; (2) inability to extract representations from arbitrary subsets of views; (3) computational costs exponentially increasing with the number of views. To address these issues, this paper proposes a scalable subset view representation learning framework based on contrastive learning. This framework reduces the number of encoder parameters by simplifying the existing encoder structure, thus lowering computational costs. It is capable of learning each subset view representation through a single objective function, making the computational cost grow linearly with the number of views. Finally, this paper introduces an operational lower bound to effectively calibrate the subset view representations based on contrastive learning. Experimental results demonstrate that this method can effectively handle various combinations of views across multiple benchmark datasets and outperforms other strong baseline methods.

**Strengths:**

1.This paper is well-structured, with clear explanations of the problems with existing methods, the proposed solutions, and the experimental results.

2.The motivation  to extract representation from an arbitrary subset of views  is commendable.

3.CwA proposes a scalable subset view representation learning framework based on contrastive learning. This framework serves as a new paradigm for multi-view representation learning, offering new insights into how different data views can enhance the model's learning capabilities.

**Weaknesses:**

1.Please provide a detailed explanation of the relationship between the analysis method presented in this paper and the one in [1], and if possible, offer a comparison with [1].

2.This paper mainly addresses two issues: scalability and calibration. However, the experimental section does not perform calibration verification.

[1] Variational Distillation for Multi-View Learning, IEEE Transactions on Pattern Analysis and Machine Intelligence, 2023, p 4551 - 4566.

**Questions:**

See the above weaknesses.

---

> ### Author Response · Authors · 2024-11-21
> **Rebuttal by Authors**
>
> Thank you for your positive feedback.
> We have addressed the comments on Weaknesses (W) as outlined below.
>
> **W1: Relationship between the proposed method and [1]**\
> Thank you for highlighting the relevance of [1].
> While both [1] and our work aim to learn representations robust to missing views using information-theoretic objectives, they differ in the following key aspects:
>
> 1. **Use of labels**: [1] directly incorporates label information during training to isolate task-relevant information in the learned representation.
>    In contrast, our method does not utilize labels or task-specific information, as we assume no prior access to such additional data.
>    This makes our approach more general and applicable to scenarios where task-relevant labels are unavailable.
>
> 2. **Treatment of view-specific information**: [1] discards view-specific information, assuming it is irrelevant to the task at hand. On the other hand, our method seeks to capture all factors of variation across views, as we do not assume prior knowledge of what information will be necessary for unknown downstream tasks.
>    This ensures our representations retain both shared and view-specific information, making them broadly applicable to a wide range of potential tasks.
>
> **W2: Calibration verification**\
> We would like to clarify that our evaluation of subset-view representations inherently assesses their calibration.
> Specifically, we evaluate each subset-view representation by using it as input to a downstream task model (e.g., classifier or regressor) that has been trained solely on complete-view representations.
> For a subset-view representation to generalize well in these downstream tasks, it must be well-calibrated toward the complete-view representation.
> Therefore, the strong performance of our method in handling subset views, as demonstrated in our experimental results, implicitly validates the calibration of these representations.
>
> If you have any suggestions for additional calibration-specific metrics or experiments, we would be happy to consider them.
>
> **Reference**\
> [1] Tian et al., Variational Distillation for Multi-View Learning, TPAMI 2023.

---

> > ### Comment · Reviewer_48cg · 2024-12-03
> >
> > Thank you for the rebuttal. I appreciate the authors' efforts in addressing my concerns. Considering the contributions of this work, I will maintain my score. However, after reviewing comments from other reviewers, I will lower my confidence score.

---

> > > ### Author Response · Authors · 2024-12-03
> > >
> > > Dear Reviewer 48cg,
> > >
> > > Thank you for your positive feedback and for highlighting the contributions of our work.\
> > > We remain committed to promptly addressing any remaining concerns raised by other reviewers.
> > >
> > > Best regards,\
> > > The Authors

---

> ### Author Response · Authors · 2024-11-29
> **Looking forward to your feedback**
>
> Dear Reviewer 48cg,\
> Thank you once again for your encouraging and thoughtful feedback.
>
> We would like to kindly remind you that we have clarified:
> 1. The relationship between our work and [1].
> 2. The calibration verification.
>
> In addition to these clarifications, we have also addressed issues raised by other reviewers, which we hope you will have an opportunity to consider as well.
>
> Our revised manuscript includes **new results**, such as the analysis of **what is being learned and ignored** in the representation (Section 4.1) and the experiment on **ImageNet-100** (Section E.6). These updates, along with our responses to other reviewers, are highlighted in the updated document for your reference.
>
> We look forward to your response and would be happy to address any further comments or questions you may have.
>
> Best regards,\
> The Authors

---

### Official Review · Reviewer_hoNY · 2024-11-02

**Soundness:** 2
**Presentation:** 3
**Contribution:** 2
**Rating:** 5
**Confidence:** 3

**Summary:**

This paper introduces Contrast with Aggregation (CwA), a scalable Multi-View Representation Learning (MVRL) framework designed to efficiently learn joint representations from diverse views. By leveraging contrastive learning with the InfoNCE objective and encoding subsets of views through Weighted Mixture of Experts (WMoE), CwA enables linear computational scaling relative to the number of views, allowing it to extract robust representations from any subset of views.

**Strengths:**

1. Compared to traditional methods, this approach demonstrates greater adaptability and scalability across different combinations of views.
2. By simplifying multiple terms that maximize mutual information (MI) into a single term, this strategy reduces computational costs, making it more effective in multi-view scenarios.
3. The paper includes open-source code, which enhances reproducibility.

**Weaknesses:**

1. The paper lacks sufficient theoretical justification for the choice of mutual information estimators and the use of specific distance metrics (Mahalanobis distance). More detailed theoretical explanation for this selection should be provided.
2. In the experiments, the paper notes that the method performs slightly worse than other baseline methods in single-view scenarios. While this issue improves with multiple view combinations, a thorough exploration of why CwA fails to surpass other methods in single-view cases and the implications for practical applications is necessary.
3. Although the paper conducts extensive experiments across various tasks, the selected datasets and task types are relatively narrow, primarily focusing on sarcasm and emotion recognition. Validation on more diverse datasets (e.g., ImageNet and NYU-Depth-V2) and tasks (e.g., action recognition on videos) should be explored to verify the method's effectiveness.
4. For the used datasets, the current advanced pretrained deep backbones should be employed to perform feature extraction for different data views instead of the somewhat handcrafted preprocessed features. More thorough studies are necessary to show its superiority.

**Questions:**

Please refer to the weaknesses part.

---

> ### Author Response · Authors · 2024-11-21
> **Rebuttal by Authors (1/2)**
>
> We appreciate your constructive review.
> We have addressed the comments on Weaknesses (W) as outlined below.
>
> **W1: Choice of Mutual Information (MI) estimator**\
> We adopted the InfoNCE objective as the MI lower-bound estimator for two primary reasons:
> - **Low variance and numerical stability**: InfoNCE is widely recognized for its low variance in MI estimation.
> Other MI lower-bound estimators [1,2,3] often suffer from the high variance of the exponential of their critic function outputs (e.g., $e^{f(z, v_{1:N})}$ in our context).
> In contrast, InfoNCE applies a softmax operation to the critic's output, ensuring that the value inside the log in Equation (7) is bounded between 0 and 1.
> This leads to reduced variance and improved numerical stability.
>
> - **Fair comparisons with CMC**: CMC also utilizes the InfoNCE objective to lower bound and maximize their MI objective between every pair of single-view representations.
> By adopting the same MI estimator, we ensure that our comparison to CMC isolates the effect of optimizing different MI objectives rather than the influence of different MI estimator choices.
>
> **W1: Choice of distance metric**\
> We chose the (squared) Mahalanobis distance as the critic function for our InfoNCE objective because it naturally aligns with our use of multivariate Gaussian representations.
> Specifically, the Mahalanobis distance measures how far the subset-view representations are from the complete-view representations, accounting for the variance of the complete-view representation.
> This ensures that each dimension is scaled appropriately based on its variance.
>
> While cosine similarity [4,5,6,7] and L2 distance [8,9] are popular choices for critic functions, they are not suitable for our representations.
> Cosine similarity is not applicable because our representations are not L2-normalized, and L2 distance is less desirable as it assumes equal scaling across all dimensions, which is not the case for our multivariate Gaussian representations.
>
> We will revise Section 3.3 to elaborate on the rationale behind selecting the InfoNCE objective and Mahalanobis distance as the critic function.
>
> **W2: Single-view performance**\
> Firstly, while our method shows lower performance on single views in the Synthetic and MOSEI datasets, it tends to outperform baseline methods with single views in the remaining datasets, as demonstrated in Tables 1, 2, and Figure 4, as well as in the additional experiment on ImageNet-100.
>
> The reason our method may be less effective with single views lies in its design.
> Our approach focuses on jointly learning representations for all combinations of views, rather than optimizing specifically for single views or particular subsets of views.
> This enables our method to perform robustly across any combination of available views, but this may also incur a small performance loss in single views.
> In contrast, most baseline methods prioritize learning representations for specific subsets, such as single views (e.g., CMC, GMC(s), and MVTCAE) or complete views (e.g., GMC(s) and MVTCAE), which limits their ability to effectively aggregate information from multiple available views, but may perform well in single views.
>
>
> ===== Updated equation index =====\
> We kindly notify you that Equation (7) referenced in our response to W1 (Choice of MI estimator) is now **Equation (4)** in the updated manuscript.

---

> ### Author Response · Authors · 2024-11-21
> **Rebuttal by Authors (2/2)**
>
> **W3: Validation on more diverse datasets**\
> We conducted additional experiments on the ImageNet-100 dataset, which consists of 100 subsampled classes selected by CMC from the full ImageNet dataset.
> Following the experimental setup in CMC, we transformed RGB images into two views in different color spaces (L and ab).
>
> To accommodate our limited computational resources, we used pretrained ResNet34 [4] to extract 512-dimensional deep features from each view.
> We then trained each representation learning method with MLP encoders to learn 128-dimensional representations from these features.
> For evaluation, we followed the same linear classification protocol as in Sections 4.2 and 4.3, training a linear classifier to predict image labels from the frozen representations.
>
> The table below summarizes the results. The results demonstrate that all methods, including ours, benefit from the addition of views.
> Notably, our method achieves superior performance compared to all baseline methods for both single-view inputs (L, ab) and two-view inputs (Lab).
> These findings indicate that our method can effectively learn representations from realistic datasets, providing further validation of its generalizability and effectiveness.
>
> | Method        | L             | AB             | LAB            |
> | ------------- | ------------- | -------------- | -------------- |
> | **CMC**       | 81.50±0.05    | 71.12±0.04     | 83.8±0.04      |
> | **GMC**       | 81.59±0.07    | 71.05±0.11     | 83.20±0.13     |
> | **GMCs**      | 80.10±0.05    | 69.62±0.13     | 82.55±0.09     |
> | **MoPoE**     | 80.30±0.04    | 68.69±0.09     | 81.66±0.14     |
> | **MVTCAE**    | 81.32±0.12    | 70.12±0.18     | 83.13±0.17     |
> | **CwA(Ours)** | **82.1±0.07** | **71.75±0.06** | **84.75±0.06** |
>
> We appreciate your suggestion and will include this additional experiment in our revised manuscript.
>
>
> **W4: Use of advanced pretrained backbones**\
> We would like to clarify that we employed pretrained deep features for feature extraction in our experiments.
> For instance, on the MUSTARD dataset (Section 4.2), we used BERT for the Text modality and ResNet for the Video modality.
>
> Additionally, with the exception of the synthetic dataset we generated, all the datasets used in our experiments are standard multi-view benchmarks that have been widely utilized in prior studies [11,12,13, 14].
> Given our focus on evaluating the representation learning methods themselves rather than the impact of backbone architectures, we prioritized following standardized preprocessing to enable a direct comparison with existing methods.
>
> We leave exploring more advanced backbones for feature extraction in future work.
>
>
> **References**\
> [1] Nowozin et al., f-gan: Training generative neural samplers using variational divergence minimization, NeurIPS 2016.\
> [2] Belghazi et al., Mutual information neural estimation, ICML 2018.\
> [3] Poole et al., On variational bounds of mutual information, ICML 2019.\
> [4] Tian et al., Contrastive multiview coding, ECCV 2020.\
> [5] Poklukar et al., Geometric multimodal contrastive representation learning, ICML 2022.\
> [6] Redford et al., Learning transferable visual models from natural language supervision, ICML 2021.\
> [7] Cherti et al., Reproducible scaling laws for contrastive language-image learning, CVPR 2023.\
> [8] Wang et al., Rethinking minimal sufficient representation in contrastive learning, CVPR 2022.\
> [9] Grill et al., Bootstrap your own latent: A new approach to self-supervised Learning, NeurIPS 2020.\
> [10] He et al., Deep Residual Learning for Image Recognition, CVPR 2016.\
> [11] Liang et al., Multibench: Multiscale benchmarks for multimodal representation learning, NeurIPS 2021.\
> [12] Liang et al., Factorized contrastive learning: going beyond multi-view redundancy, NeurIPS2023.\
> [13] Hwang et al., Multi-view representation learning via total correlation objective, NeurIPS 2021.\
> [14] Li et al., Large-scale multi-view spectral clustering via bipartite graph, AAAI 2015.

---

> ### Author Response · Authors · 2024-11-29
> **Looking forward to your feedback**
>
> Dear Reviewer hoNY,\
> Thank you once again for your thoughtful and constructive feedback.
>
> We have worked diligently to address your concerns, specifically:
>
> 1. Providing a justification for our choice of MI estimator and distance metric.
> 2. Discussing the observed single-view performance.
> 3. Conducting additional experiments on ImageNet-100.
> 4. Clarifying the usage of deep pretrained backbone encoders.
>
> In addition, we have addressed issues raised by other reviewers, which we hope you may find relevant as well.
>
> Our revised manuscript includes **new results**, such as the analysis of **what is being learned and ignored** in the representation (Section 4.1) and the experiment on **ImageNet-100** (Section E.6). These updates, along with our detailed responses to the reviews, are highlighted in the document for your reference.
>
> We would greatly appreciate your further feedback and would be happy to address any additional comments or questions you may have.
>
> Best regards,\
> The Authors

---

### Official Review · Reviewer_Vxpo · 2024-11-02

**Soundness:** 2
**Presentation:** 3
**Contribution:** 2
**Rating:** 6
**Confidence:** 4

**Summary:**

This paper presents Contrast with Aggregation (CwA) framework for multi-view representation learning that efficiently aggregates information from any subset of views using contrastive learning. CwA addresses computational and robustness challenges in multi-view learning by leveraging an information-theoretic objective and Mahalanobis distance to jointly train subset-view representations with a computation cost that scales linearly with the number of views. Extensive experiments demonstrate CwA's effectiveness and scalability, showing it outperforms baseline methods across multiple benchmark datasets source.

**Strengths:**

(1) The paper tackles computational challenges in multi-view learning by using an information-theoretic objective and Mahalanobis distance to train subset-view representations jointly, achieving a computation cost that scales linearly with the number of views.

(2) The paper has presented extensive experiments showing the effectiveness and scalability of CwA. The paper is clearly written and different details are well understood.

**Weaknesses:**

(1) The paper lacks an explicit statement of its contributions, leaving its main claims unclear. This should definitely be clarified in the introduction. Furthermore, the related works section does not sufficiently situate the proposed approach within the existing body of research. Without these clarifications, it is challenging to fully assess the paper.

(2) The paper could be seen as an extensive application of the inverse variance weighted (IVW) average for information fusion. However, it does not investigate whether this fusion method is optimal for various downstream tasks. Additionally, it would be valuable to explore whether this fusion approach is effective when applied with other metrics as well.

(3) Table 1 indicates that, for some datasets, performance achieved with a single modality surpasses that with combined views. For instance, on the MOSI dataset, CwA attains a classification accuracy of 67.23% using only the text modality, whereas combining all three views results in a lower accuracy of 65.61%, suggesting that information fusion may actually reduce model performance. This observation and a reasoned explanation is not discussed in the paper. So, I am looking for some insightful discussion to understand the reasons behind this observation.

**Questions:**

Please see the weaknesses section

---

> ### Author Response · Authors · 2024-11-21
> **Rebuttal by Authors (1/2)**
>
> Thank you for your insightful review.
> We have addressed the comments on Weaknesses (W) as outlined below.
>
> **W1: Clarification on main claims**\
> We have outlined our contributions more explicitly as follows:
>
> 1. **Theoretical contribution**: Proposition 1 in Section 3.2 formally demonstrates that the single MI term between complete views and the joint representation encoded by any WMoE joint encoders (e.g., MoE, PoE, MoPoE) provides a lower bound of the weighted average of various MI terms. This enables learning representations for various subsets of views efficiently.
>
> 2. **Algorithmic contribution**: In Section 3.3, we derived a tractable lower bound for our single MI objective, enabling the MoPoE joint encoder to learn and calibrate exponentially many subset-view representations with a computational cost that scales linearly with the number of views. Importantly, this represents a significant improvement over previous work [1], which trained the same encoder with a computational cost that increased exponentially with the number of views.
>
> 3. **Empirical contribution**: Through comprehensive evaluations on seven MVRL benchmark datasets, including newly added results on ImageNet-100 (as presented in our response to reviewer hoNY), spanning 2 to 8 views, we demonstrated that our method achieves robust performance across diverse input view combinations, consistently surpassing strong baseline methods.
>
> We will revise the introduction section to explicitly highlight these contributions to ensure clarity.
>
> **W1: Situating our work within related work**\
> Thank you for pointing this out. To address this concern, we would like to note that our method is a Contrastive MVRL approach that builds upon and optimizes the encoders originally introduced by Multi-View VAEs. As such, it is closely related to prior work in both Contrastive MVRL and Multi-View VAEs. We will revise Section 2 to explicitly highlight these connections.
>
> If you have any additional suggestions for better situating our work within the context of related research, we would greatly appreciate your input.
>
> **W2: On the optimality of IVW fusion**\
> Inverse-Variance Weighted (IVW) average [2, 3, 5] is a classical (late) fusion method widely used in statistical sensor fusion.
> When observations are generated from the underlying state with independent Gaussian random noise, the IVW average serves as the Maximum Likelihood Estimation (MLE) of the state.
> To assess its optimality within the context of MVRL, let $\mu$ represent the ground-truth state that encapsulates all factors of variation for a given instance.
> Assuming that the mean of each single-view representation $\mu_i$ from the *i*-th view is an observation of $\mu$ with Gaussian noise $\sigma^2_i$ s.t. $p(\mu_i | \mu) = N(\mu, \sigma^2_i \text{I})$, the IVW average $\mu_{1:N} = \frac{\sum_{i=1}^{N} \mu_i / \sigma^2_i}{\sum_{i=1}^{N} 1 / \sigma^2_i}$ is the MLE of $\mu$.
> This assumption aligns naturally with our method, as all 𝜇ᵢ are derived from the same instance but exhibit different precisions based on the characteristics of each view.
> These precisions are captured by the learned variance $\sigma^2_i$ of each single-view representation.
> Consequently, IVW allows each single-view representation to contribute to the complete-view representation in proportion to its precision $\frac{1}{\sigma^2_i}$, making it an optimal choice for MVRL.
>
> Beyond its statistical foundation, IVW offers computational scalability, with costs that scale linearly with the number of input views.
> This aligns with our goal of retaining scalability in MVRL.
> While alternatives like cross-view attention-based fusion could be considered, they incur a quadratic computational cost relative to the number of input views, making them less practical for scenarios involving many views.
> Moreover, such approaches have been empirically shown to perform less effectively when handling subset views, as observed in [4].
>
> If there are other fusion approaches you believe could serve as valuable comparisons, we would greatly appreciate your suggestions to explore them further.
>
> Additionally, regarding the statement about the effectiveness "when applied with other **metrics**," we kindly request clarification on the specific metrics you are referring to so that we can address this point more effectively.

---

> ### Author Response · Authors · 2024-11-21
> **Rebuttal by Authors (2/2)**
>
> **W3: Lower performance with combined views**\
> Thank you for highlighting the case of reduced performance when combining views, which is indeed an important observation.
> While this issue is specific to the Text modality on the MOSI dataset in our method, it can occur when the informativeness of modalities is highly unbalanced for the downstream task.
> Specifically, the Text modality (T) is inherently more informative for sentiment classification, as it often contains keywords that make sentiment inference trivial.
> This explains why, in Table 1, the performance of the Text modality alone is consistently higher than that of any other single-view modality across all methods.
> In such cases, combining representations from multiple views via a (weighted) average may slightly degrade the representation from the most informative view.
> This occurs because each view’s representation contributes to all dimensions of the combined representation to some extent, potentially diluting the signal from the dominant view.
> Importantly, this phenomenon is not unique to our method. For example:
>
> - CMC achieves its highest performance using the Text modality alone on MOSI and MOSEI.
> - GMC achieves its highest performance using the Text modality alone on MUSTARD.
> - Similarly, MoPoE and MVTCAE fail to improve upon the performance of the Text modality when adding Video or Audio as additional views on MUSTARD.
>
> We appreciate the opportunity to address this observation and will include a discussion of this phenomenon in the revised paper to provide further insights.
>
> **References**\
> [1] Sutter et al., Generalized multimodal elbo, ICLR 2021.\
> [2] Cochran et al., A sampling investigation of the efficiency of weighting inversely as the estimated variance, Biometrics 1953.\
> [3] Cochran et al., The combination of estimates from different experiments, Biometrics 1954.\
> [4] Hwang et al., Information-theoretic state space model for multi-view reinforcement learning, ICML 2023.\
> [5] Hinton et al., Training products of experts by minimizing contrastive divergence, Neural Computation 2002.

---

> ### Comment · Reviewer_Vxpo · 2024-11-24
>
> I thank the authors for the rebuttal, which partially resolve my concerns.
>
> 1. I think the contributions made by the paper are kind of clear now. To me, the theoretical and algorithmic contributions revolve around the MI term which acts as a lower bound of the objective. However, I think the derivation of this term is highly inspired from (Shi et al., 2019) and (Sutter et al., 2021).
>
> 2. This is now resolved from authors answers and the additional experiments on ImageNet-100 dataset that they have presented in the reply of reviewer hoNY.
>
> 3. The performance of information fusion methods on a given dataset seems to depend on the specific method-dataset combination. This highlights the importance of discussing how to select the appropriate method for a particular dataset. Additionally, it is crucial to consider how the informativeness of a specific modality is defined and what happens to that informativeness when the modality is combined with others. These are significant questions that need to be addressed in the context of any information fusion method. I think some analysis of what is being learnt and what is being ignored in the context of modality fusion should be done, which is kind of missing in the present version.

---

> > ### Author Response · Authors · 2024-11-27
> > **Response to follow-up review**
> >
> > We sincerely appreciate your follow-up review. Below, we address each aspect of the remaining concerns.
> >
> > **1: Contributions**\
> > We would like to emphasize that the derivation of Proposition 1 in Section 3.2 is our novel contribution. While our work leverages the encoder structures introduced in MMVAE (Shi et al., 2019) and MoPoE-VAE (Sutter et al., 2021), we do not borrow any derivations from their ELBO-based objectives. Instead, our approach is built on a novel MI-based formulation that is entirely distinct from the ELBO formulation. This distinction allows our method to consistently outperform MoPoE-VAE on most input view combinations across all datasets (except Caltech-101) while significantly reducing the computational cost of training the encoders.
> >
> > Moreover, adopting the MoPoE joint encoder in a contrastive learning (CL) objective is an unexplored area in prior work. We believe that our formulation, which combines the MoPoE framework with CL from an information-theoretic perspective, provides a significant contribution to the field of MVRL. This point has also been positively noted by Reviewer 48cg.
> >
> > **3: Method-dataset selection \& modality informativeness**\
> > Selecting the optimal method for a dataset or defining the informativeness of each modality is inherently ill-posed because both heavily depend on the unknown downstream task rather than the dataset itself. For instance, consider the MOSI dataset in MultiBench. While the evaluation is performed on the sentiment classification task specified as part of the dataset (e.g. sentiment classification for MOSI and emotion classification for MOSEI), the actual downstream task could be anything. For example, the task could be the gender classification of the speaker. In this scenario, there is no guarantee that methods found to perform well for sentiment classification generalize well to the gender classification task, as the tasks require different types of information. Similarly, while the text modality is highly informative for sentiment classification, it becomes less useful for the gender classification task compared to video and audio modalities. This highlights the inherent challenge in MVRL, where prior knowledge about downstream tasks or modality relevance cannot be assumed.
> >
> > Our method addresses this challenge by effectively capturing both shared and view-specific factors of variation, as demonstrated in Figure 2 in the revised Section 4.1.
> > This capability ensures robustness across a wide range of downstream tasks, as also emphasized by prior works [1, 2, 3].
> >
> > **3: Analyzing what is learned \& ignored in modality fusion**\
> > We fully agree that analyzing what is being captured or ignored in the representation is essential. However, this type of analysis is particularly challenging with in-the-wild datasets like MOSI, MUSTARD, and others due to the lack of controlled variables. To address this, we conducted such analysis on the Synthetic dataset, where all data-generative factors are fully known. Specifically, we measured two types of Mean Squared Errors (MSE):
> > 1. The MSE between the true **shared** generative factor $g_s$ and its predicted value from $z$ as input views are accumulated (e.g., view 1, views 1+2, ..., views 1:N) in the test set.
> > 2. The MSE between the true **view-specific** generative factors $[g_1,...,g_8]$ and their predicted values from the same $z$.
> >
> > These analyses allow us to explicitly evaluate what aspects of the data are being captured in the learned representations. To highlight these findings, we revised our manuscript by including these results in Figure 2 and Section 4.1, while moving the previous results on the joint prediction of shared and view-specific factors $[g_1,...,g_8]$ to Section E.7 of the supplementary material.
> >
> > Figure 2(a) and Figure 2(b) show that only CwA effectively aggregates view-specific factors of variation while preserving the ability to capture shared factors, unlike baseline methods, which tend to discard view-specific factors. This distinction is further detailed in the results paragraph of Section 4.1 in the revised manuscript.
> >
> > **References**\
> > [1] Hwang et al., Multi-view representation learning via total correlation objective, NeurIPS 2021.\
> > [2] Lee et al., Image representation using 2d gabor wavelets, IEEE TPAMI 1996.\
> > [3] Zhang et al., CPM-Nets: cross partial multi-view networks, NeurIPS 2019.

---

> ### Author Response · Authors · 2024-11-29
> **Looking forward to your additional feedback**
>
> Dear Reviewer Vxpo,\
> Thank you once again for your thoughtful follow-up review.
>
> As the author-reviewer discussion period is set to conclude in a few days, we would greatly appreciate your feedback on whether our response and the revised manuscript have effectively addressed your concerns.
> If there are any remaining comments or questions, we would be more than happy to address them.
>
> Thank you for your valuable time and input.
>
> Best regards,\
> The Authors

---

### Official Review · Reviewer_zRh5 · 2024-11-03

**Soundness:** 2
**Presentation:** 1
**Contribution:** 2
**Rating:** 5
**Confidence:** 4

**Summary:**

This paper proposes a scalable multi-view representation learning (MVRL) framework named CwA based on contrastive learning to address the challenges in MVRL. It can effectively aggregate information from any subset of views. It optimizes each subset-view representation with a computational cost that increases linearly with the number of views, thus optimizing the computational cost. In addition, by integrating the Mahalanobis distance into the InfoNCE objective, this article reformulates their method as a contrastive learning (CL) approach for calibrating each subset-view representation. It has achieved good results on multiple datasets.

**Strengths:**

They introduced CwA, a scalable Multi View Representation Learning (MVRL) framework that can effectively aggregate information from any subset of views. Not only did it reduce the complexity of some models, decrease the complexity of parameter count and training time, but it also achieved better results than most related works.

**Weaknesses:**

1.This paper describes the reduction in the number of encoder parameters, but lacks a comparison of the number of encoder parameters in the paper. The effectiveness and necessity of the paper's method cannot be verified through poor experiments.
2.The explanation of related work in the paper is relatively simple, and it is relatively difficult for non professionals to understand the paper.  Besides, the role of the encoder and the significance of the operations performed in this article are not clear enough, and the description of the methods section is not concise enough. A large amount of space is provided to supplement the explanation, which I think is unreasonable.
3. This article proposes that the model can be modal extended, but I did not see any difference in this scalability compared to previous models in the paper. Please clarify with the submitter.

**Questions:**

1.Have you ever considered why the model performs relatively poorly in a single modality? Is it due to hyperparameters?
2.Are the results obtained by other methods tested by the author themselves on the server or published in other papers? Does the author think it has any reference value?
3.Has the author considered the performance of the model in subsequent tasks?

---

> ### Comment · Reviewer_zRh5 · 2024-11-14
> **According to the feedback from the Associate Program Chairs, the reviews are further refined**
>
> 1. Their paper states in section 3.1 that the parameter count and computational cost of the encoder increase relatively less with increasing views compared to other models. If it is a deep learning model, can the network weight parameter quantity or iteration time be provided for comparison, which will further prove its viewpoint. In addition, in the ablation experiment of computational cost, in the case of relatively few views, the difference in computational cost among multiple models is not significant, and even the computation time of CwA is longer.
>
>
> 2. The explanation of the relevant work in the paper is relatively simple, appearing to simply cite a large number of references rather than introducing methods, and it is not clear how the multi-view VAE methods are related to the method proposed in this paper. In addition, the explanation of the role of the encoder and the significance of the operations performed in this paper is not clear enough.
>
>
> 3. This paper proposes that the model can handle multi-view extensions, but I feel that this has not been fully validated from the experimental part. The experiment is divided into three parts: synthetic dataset, multimodal dataset, and conventional multi-view dataset. I am not sure if the synthetic dataset has practical significance, and how to discuss the effectiveness of the simulation experiment? The multimodal dataset used by the author lacks detailed explanations, and only four comparison methods were selected, lacking more state-of-the-art methods for comparative experiments.

---

> ### Author Response · Authors · 2024-11-21
> **Rebuttal by Authors (1/2)**
>
> We appreciate your valuable feedback.
> We have addressed the comments on Weaknesses (W) and Questions (Q) as outlined below.
>
> **W1: Parameter count**
>
> | Method                    | Synthetic (8 views) | 4 datasets  in MultiBench (3 views) | Caltech-101 (6 views) |
> | ------------------------- | ------------------- | --------------------------------------------------------- | --------------------- |
> | **CwA (Ours)**            | 0.8M                | 15M                                                       | 3.5M                  |
> | **CMC**                   | 0.8M                | 15M                                                       | 2.7M                  |
> | **GMCs**                  | 1.3M                | 15M                                                       | 3.6M                  |
> | **GMC**                   | 2M                  | 30M                                                       | 5.6M                  |
> | **MoPoE-VAE & MVTCAE** | 1.4M                | 17M                                                       | 6.2M                  |
>
> We have summarized the number of parameters used by each method across all datasets in the table above.
> It is important to note that all the methods compared in this work are late-fusion algorithms that use per-view backbone encoders with the same number of parameters for preprocessing input views.
> Consequently, the total number of model parameters for all methods scales linearly with the number of views, as each additional view introduces a fixed number of parameters.
> For example, doubling or tripling the number of input views simply doubles or triples the parameter count for all methods.
> However, the specific design choices in processing features encoded by backbone encoders and additional network components (e.g., decoders or extra backbone encoders) result in differences in the overall parameter count, which we elaborate on below:
>
> **CMC** encodes features from the backbone encoders into deterministic single-view representations using one linear layer per view.
> In comparison, **CwA** uses two linear layers to encode the features into the mean and variance of single-view representations.
> This design results in a slightly higher parameter count for CwA in Caltech-101 with small backbone encoders, while the parameter counts remain nearly identical for the rest of the bigger datasets.
>
> **GMCs** encodes backbone features into single-view and complete-view representations using two linear layers shared across all views.
> However, the linear layer for complete-view representation processes a stack of all backbone features, leading to a relatively larger parameter count in datasets like Synthetic and Caltech-101.
> **GMC** introduces an additional backbone encoder per view on top of GMCs, leading to up to twice the number of parameters compared to CMC, CwA, and GMCs.
>
> While sharing the same per-view encoder architecture as CwA, **MoPoE-VAE** and **MVTCAE** incorporate decoders, which can at most double their overall parameter count compared to other methods.
> Further details on the architectures of encoders and decoders for each dataset can be found in Tables 4, 5, 6, and 7 in Section D of the supplementary material.
>
> **W1: Iteration time**\
> Regarding iteration time, we would like to highlight that computation time becomes increasingly critical as the number of views grows, which is the primary focus of our work.
> Figure 3 in the main text reports the running time for 10 training epochs, showing that our method is relatively slower when handling a small number of views (2–4) compared to contrastive learning methods.
> This is because computing the Mahalanobis distance between representations in our method is more computationally intensive than the inner product operations used by CMC and GMC(s).
> However, when scaling to 8 or 16 views, our method outperforms all other approaches, including GMCs, GMC, and MVTCAE, even though these methods also have computation costs that scale linearly with the number of views.
>
> **W2: Shortened explanation of related work**\
> Due to space limitations, we had to keep the discussion of early-fusion approaches in Section 2 brief.
> However, we provided an in-depth explanation of late-fusion approaches, including Contrastive MVRL methods and Multi-View VAEs, as our method is a Contrastive MVRL approach that builds on and optimizes the encoders introduced by Multi-View VAEs.
> This makes our method closely related to these works.
> For a more detailed explanation of the relationships between our method and these late-fusion approaches, we direct the reviewer to Section B in the supplementary material.
>
> **W2: Unclear explanation of the encoder and the operations**\
> Regarding the explanation of the encoder and the operations performed in this paper, we kindly ask the reviewer to specify the unclear parts (e.g., by referencing line numbers), which would allow us to provide a more precise and helpful clarification.

---

> ### Author Response · Authors · 2024-11-21
> **Rebuttal by Authors (2/2)**
>
> **W3: The purpose and the significance of the synthetic dataset**\
> The purpose of using the synthetic dataset was two-fold: (1) to ensure the presence of both a shared factor observable across all views and view-specific factors unique to each view, and (2) to evaluate whether each MVRL method can effectively capture both types of information.
> The synthetic dataset is generated with 9 disentangled factors: 1 shared factor and 8 view-specific factors across 8 views.
> Regression on these factors as a downstream task provides a controlled way to evaluate whether a given MVRL method captures all important information in its representations.
> While the synthetic dataset may lack direct practical applicability, we believe that its ability to offer a controlled setup for verifying key capabilities (e.g., aggregating complementary information across views) makes it valuable for tasks with real-world analogs, such as autonomous driving, medical diagnostics, or multi-sensor systems.
>
> In contrast, tasks from other datasets tend to focus more on shared factors. For example, in the four datasets from MultiBench, sentiment is often inferable from any single modality (video, audio, or text), with text being more advantageous when it contains critical keywords.
> Similarly, in Caltech-101, the identity of each object is inferable from any of its six visual features.
>
> **W3: Explanation on multimodal datasets**\
> Thank you for pointing this out. We will revise Section D.2 in the supplementary material to provide the following information about the datasets:
>
> The four multimodal datasets (MOSI, MUSTARD, FUNNY, and MOSEI) were collected to explore human affective states through diverse expressions, including spoken language, facial expressions, gestures, and speech tone.
> These datasets represent human expressions as multimodal time-series data comprising text, video, and audio modalities, enabling the prediction of sentiment (MOSI), emotion (MOSEI), humor (FUNNY), and sarcasm (MUSTARD).
> These modalities provide complementary information, highlighting the importance of understanding their complex relationships. Further details on each dataset are available in [1].
>
> **W3: Comparison to state-of-the-art methods**\
> We would like to ask you to provide references to any late-fusion approaches we may have missed so that we can include them for comparison.
>
> **Q1: Single-view performance and hyperparameter influence**\
> Firstly, while our method shows lower performance on single views in the Synthetic and MOSEI datasets,
> it tends to outperform baseline methods with single views in the remaining datasets,
> as demonstrated in Tables 1, 2, and Figure 4, as well as in the additional experiment on ImageNet-100 (in the response to reviewer hoNY).
>
> The reason our method may be less effective with single views lies in its design.
> Our approach focuses on jointly learning representations for all combinations of views, rather than optimizing specifically for single views or particular subsets of views.
> This enables our method to perform robustly across any combination of available views, but this may also incur a small performance loss in single views.
> In contrast, most baseline methods prioritize learning representations for specific subsets, such as single views (e.g., CMC, GMC(s), and MVTCAE) or complete views (e.g., GMC(s) and MVTCAE), which limits their ability to effectively aggregate information from multiple available views, but may perform well in single views.
>
> Regarding hyperparameters, our method demonstrates robust performance across a wide range of input views, as evidenced by the hyperparameter sensitivity analysis provided in Section E.1. This suggests that performance is not significantly impacted by the choice of hyperparameters.
>
> **Q2: Regarding reference value**\
> Since our evaluation scenarios are not identical to the settings used in the original papers for the baseline methods, we conducted all evaluations using our own computational resources. While we do not have reference values for direct comparison, we have provided anonymized code and detailed explanations on evaluation protocol (Section 4) and implementation (Section D in the supplementary material) to ensure the correctness of our evaluations and support reproducibility.
>
> **Q3: Regarding subsequent task**\
> Could you kindly clarify what is meant by "subsequent tasks"?
>
> **Reference**\
> [1] Liang et al., Multibench: Multiscale benchmarks for multimodal representation learning, NeurIPS 2021.

---

> > ### Comment · Reviewer_zRh5 · 2024-11-25
> >
> > I thank the authors for the rebuttal and revised paper, which to some extent resolved my doubts.
> >
> > Through the author's explanation, I gradually understood the significance of multi view VAEs in related work. Meanwhile, the experiments on the number of parameters and iteration time also fully answered my review.
> >
> > However, I still have some problems that prompted me to maintain my initial rating  5: marginally below the acceptance threshold.
> > (1) In the revised paper, the empirical contribution said the authors utilized 7 MVRL benchmark datasets, but I only find six datasets: one synthetic dataset, four multimodal datasets and one normal multi-view dataset.
> >
> > (2) What puzzles me the most is the authors did not analyze why their proposed CwA model performed much better on synthetic dataset than existing comparison methods, but showed inconsistent performance on the four multimodal datasets and Caltech101 dataset.  In the authors' rebuttal, it explains the differences between the synthetic dataset and other datasets, but it cannot be determined whether the results of the synthetic dataset have reference value. Inconsistent performances are always confusing.
> >
> >
> > （3）From Table 1, the comparison results show confusion. On the MUSTARD dataset, when GMC only uses the text view, the accuracy rate is 64.49%, which is higher than the author's method's 64.28% across three views. Meanwhile, a strange phenomenon is that on the MOSI dataset, using one or two views of data performs better than using three views, which is inconsistent with the author's claim of integrating information from multiple views. Why spend more cost to obtain multiple view data if one view data is sufficient to achieve better results？ In addition, in Table 1, the author's method presents ideal results for the MOSEI dataset, where the more views there are, the more significant the results are. The four datasets have different data volumes and modal dimensions, and the author did not provide sufficient analysis of these influencing factors.
> >
> >
> > (4) On the common multi-view dataset Caltech101, the authors' CwA method performed worse than existing VAE methods. Therefore, the author designed a new CwA+recon method, which yielded results far superior to the comparison methods. It is puzzling that this new method has not been proposed on other datasets.
> >
> >
> > Overall, existing experiments and analysis cannot support me in improving my rating

---

> ### Author Response · Authors · 2024-11-27
> **Response to follow-up review (1/2)**
>
> We sincerely appreciate your follow-up review. Below, we address each aspect of the remaining concerns.
>
> **1: 7 MVRL benchmark datasets**\
> In addition to the six datasets that were originally in the paper, we conducted additional experiments on ImageNet-100, as requested by Reviewer hoNY. The results from the new dataset are reported in Section E.6 of the revised manuscript and also discussed in our response to Reviewer hoNY.
>
> **2: Inconsistent performances**\
> The main purpose of the synthetic dataset experiments is to complement the standard benchmark datasets (i.e. MultiBench, Caltech-101, ImageNet) commonly used in the MVRL literature. Since these datasets are focused on measuring the task-specific performance, it is not clear that the baseline methods are successfully capturing view-specific factors even when they are performing well on the downstream task. The complexity of the data (image, text, video, etc.) also hinders an in-depth analysis. This is an on-going problem with all the evaluations appearing in the prior literature on MVRL, as they don’t show the whole picture. In order to raise awareness about the issue for the community, we have proposed the synthetic dataset that allows us to directly measure whether the method is successfully capturing shared as well as view-specific factors, shown in figure 2 (a) and (b) in the revision.
>
> As shown in figure 2 (b), all the baseline methods fail to encode view-specific factors. This is because our method is designed to successfully aggregate both shared and view-specific factors, while other methods tend to discard view-specific factors. This distinction is discussed in detail in the results paragraph of Section 4.1 in the revision.
>
> The apparent ‘inconsistent’ performance in MultiBench and Caltech-101 is because, as we stated earlier, it only measures the downstream task performance. Throwing away view-specific factors, such as the video-specific information that is irrelevant to the downstream task (such as the gender of the speaker for the sentiment classification) will not affect the performance, and as such, all the baseline methods tend to do so as observable from the synthetic dataset experiments. The takeaway from these experiments is that, CwA doesn’t fall into this common trap, and in addition, the shared factors found by CwA are generally better than those found by other baselines, albeit it doesn’t completely dominate over all datasets. It is important to note that it is extremely challenging for a single method to achieve superior performance across all input-view combinations and benchmark datasets, as evidenced by the evaluation results of other baseline methods [1, 2, 3].
>
> Lastly, as emphasized in prior works [3, 4, 5], learning complete representations that capture shared as well as view-specific factors of variation is crucial for MVRL because we cannot assume prior knowledge about downstream tasks or which information will be most relevant. This highlights the importance of our synthetic dataset, which allows us to rigorously assess whether a method captures all factors of variation. We are unable to provide reference values, as the synthetic dataset experiment is a novel contribution of our work. The results presented in our paper serve as the reference values.
>
> **References**\
> [1] Tian et al., Contrastive multiview coding, ECCV 2020.\
> [2] Sutter et al., Generalized multimodal elbo, ICLR 2021.\
> [3] Hwang et al., Multi-view representation learning via total correlation objective, NeurIPS 2021.\
> [4] Lee et al., Image representation using 2d gabor wavelets, IEEE TPAMI 1996.\
> [5] Zhang et al., CPM-Nets: cross partial multi-view networks, NeurIPS 2019.\

---

> ### Author Response · Authors · 2024-11-27
> **Response to follow-up review (2/2)**
>
> **3: Performance of GMC with single text view**\
> Regarding the performance of GMC with the single text view, its use of two backbone encoders per view provides a significant advantage, particularly given the high dimensionality of the text view in the MUSTARD dataset (50x300 dimensions). This is supported by the results of GMCs, which employs a single backbone encoder per view (indicated by "s" for single backbone) for fair comparison, which consistently underperforms compared to our method across all input view combinations.
>
> **3: Lower performance with additional views on MOSI**\
> Regarding the lower performance with additional views on the MOSI dataset, we reiterate our response to Reviewer Vxpo:
>
> This phenomenon is specific to the Text view on the MOSI dataset. More generally, this can occur when the informativeness of views is highly unbalanced for the downstream task. For example, the Text view (T) is inherently more informative for the downstream task (i.e. sentiment classification), as it often contains keywords that make it trivial to infer the sentiment. This explains why, in Table 1, the performance of the Text view alone is consistently higher than that of any other single views across all methods. In such cases, combining representations from multiple views via a (weighted) average may slightly degrade the representation from the most informative view. This occurs because each view’s representation contributes to all dimensions of the combined representation to some extent, potentially diluting the signal from the dominant view. More importantly, this phenomenon is not unique to our method. For example:
> - CMC achieves its highest performance using the Text view alone on MOSI and MOSEI.
> - GMC achieves its highest performance using the Text view alone on MUSTARD.
> - Similarly, MoPoE and MVTCAE fail to improve upon the performance of the Text view when adding Video or Audio as additional views on MUSTARD.
>
> We included our discussion on this issue in Section 4.2 of our revised manuscript.
>
> **3: Analysis of influencing factors**\
> Please understand that analyzing performance on in-the-wild datasets, such as MOSI, MUSTARD, and others, is inherently challenging due to the lack of controlled variables. This is one of the primary motivations for including our synthetic dataset, where all influential factors can be explicitly designed and evaluated consistently across all methods.
>
> **4: CwA+recon**\
> Thank you for raising this question. When input views are well feature-engineered to represent only the essential characteristics of an instance, autoencoder-based approaches can be advantageous, as the reconstruction objective encourages the representation to memorize the views. For this reason, we evaluated CwA+recon on Caltech-101, which consists of six handcrafted visual feature views, to explore whether our method could benefit from the reconstruction objective in such scenarios.
>
> However, in-the-wild datasets in practice consist of high-dimensional data with sparse and redundant information. For example, the video, audio, and text views in the four datasets from MultiBench typically exhibit sequential structures, making them high-dimensional and containing overlapping information across adjacent timesteps. In such cases, forcing the representation to memorize every detail through autoencoding may not be beneficial and can hinder the discovery of meaningful global context information. This is reflected in our results on MultiBench (Tables 1 and 2), where autoencoder-based methods such as MoPoE-VAE and MVTCAE generally underperform compared to contrastive learning (CL) methods, especially on MUSTARD and MOSEI. In these scenarios, incorporating an autoencoder objective into our method is unlikely to provide additional benefits.
>
> A similar trend is evident in our new results on ImageNet-100, where CL-based methods remain more effective than Multi-View VAEs. For these reasons, we opted not to evaluate CwA+recon on datasets with high-dimensional, in-the-wild views.

---

> ### Author Response · Authors · 2024-11-29
> **Looking forward to your additional feedback**
>
> Dear Reviewer zRh5,\
> Thank you again for your thoughtful follow-up review.
>
> With the author-reviewer discussion period coming to a close in a few days, we would sincerely appreciate your feedback on whether our response and the revised manuscript have sufficiently addressed your concerns.
>
> Additionally, we would like to highlight some **new results** included in the revised manuscript, such as the analysis of **what is being learned and ignored** in the representation (Section 4.1) and the experiment on **ImageNet-100** (Section E.6), which we hope you find relevant.
>
> If there are any remaining questions or additional comments, we would be happy to address them promptly.
>
> Thank you for your valuable time and input.
>
> Best regards,\
> The Authors

---

### Author Response · Authors · 2024-11-21
**Rebuttal by Authors**

# General Response

We sincerely appreciate the reviewers for their constructive and thoughtful feedback, which have greatly contributed to improving the quality of this work.

In our responses to the reviews, we have carefully addressed all raised concerns. These can be summarized as follows:

- **Additional experiments** (hoNY): We conducted experiments on ImageNet-100 to further validate our method and provided results demonstrating its effectiveness in realistic data.
- **Theoretical justification** (hoNY): We elaborated on the rationale for selecting the InfoNCE objective as the MI estimator and the Mahalanobis distance as the critic function, emphasizing their suitability and advantages in our context.
- **Optimality of IVW fusion** (Vxpo): We provided a detailed explanation on the inverse variance-weighted (IVW) average, showing its statistical grounding and addressing concerns regarding its optimality.
- **Discussion on single-view performance** (zRh5, hoNY): We discussed why our method performs relatively weakly with single views, while highlighting its robustness with multi-view combinations.

- **Relationship to prior work** (zRh5, Vxpo, 48cg): We clarified how our method builds upon and extends concepts introduced by Multi-View VAEs and Contrastive MVRL approaches, situating our work within the context of related research.

Due to space limitations, we are actively revising our manuscript to carefully incorporate all the valuable feedback from the reviewers. This ongoing process includes:
- Adding further explanations and clarifications to Sections 1, 2, and 3.
- Expanding experimental details, including descriptions of the four datasets from MultiBench and additional results from ImageNet-100.
- Enhancing the overall presentation to improve clarity and accessibility.

We are fully committed to completing these revisions as soon as possible and will promptly notify reviewers once the updated manuscript is available. If there are any remaining concerns or suggestions, we welcome the opportunity to address them during the ongoing discussion period.

---

### Author Response · Authors · 2024-11-24
**Updated Manuscript**

Dear Reviewers,

We sincerely appreciate your valuable feedback and have carefully incorporated your suggestions into the revised manuscript.\
The updates are highlighted in blue text for clarity.

Below is a summary of the key changes:

- Explicit statement of **contributions** in Section 1 (Vxpo).
- **Positioning of our work** within the MVRL context in Section 2 (Vxpo).
- Justification for the **InfoNCE objective and Mahalanobis distance** in Section 3.3 (hoNY).
- Detailed **explanations on the 4 datasets** from MultiBench in Section 4.2 (zRh5).
- Discussion on the **lower performance of combined views** in Section 4.2 (Vxpo).
- New experiment results on **ImageNet-100** in Section E.6 (hoNY).
- Discussion on the **optimality of IVW** within MVRL in Section G (Vxpo).

We welcome further discussion on any remaining concerns and are happy to incorporate additional suggestions to further improve our work.

Thank you for your thoughtful feedback and support throughout the review process.

---

> ### Author Response · Authors · 2024-11-27
> **Second revision**
>
> Dear Reviewers,
>
> In response to the suggestion from Reviewer **Vxpo**, we have incorporated an analysis of **what is learned and ignored** in the representation using the synthetic dataset. This analysis is detailed in the revised Section 4.1.
>
> Additionally, we updated Figure 5 to clarify the data generation process for the synthetic dataset and elaborated on the importance of this dataset in Section D.1.
>
> All updates in this second revision are highlighted in magenta.
>
> If you have any remaining concerns, we would be more than happy to address them.
>
> Thank you for your constructive feedback.

---

### Author Response · Authors · 2024-12-03
**Summary of the Rebuttal and Revision Status (D-7h)**

Dear Reviewers,

We believe that we have thoroughly addressed all comments from reviewers through our responses and the revised manuscript.\
Below, we summarize the updates made to address the issues raised:

- Conducted an analysis of what is learned and ignored in representations using the controlled synthetic dataset (Vxpo), while clarifying the significance of synthetic datasets in evaluating multi-view representation learning (zRh5).
- Performed additional experiments on ImageNet-100 as requested, demonstrating the effectiveness of our method on realistic datasets (hoNY).
- Provided a detailed explanation of the theoretical and algorithmic contributions of the method, emphasizing the novelty of its information-theoretic objective (Vxpo).
- Justified the use of IVW-based encoders, the InfoNCE objective, and Mahalanobis distance as the critic function (Vxpo, hoNY).
- Discussed challenges in defining modality informativeness and selecting the optimal method-dataset combination due to the unsupervised nature of multi-view representation learning (Vxpo).
- Addressed concerns about performance inconsistencies in MultiBench and explained how these reflect common challenges in multi-view representation learning (zRh5).

In addition, we carefully revised the manuscript with the following changes:
- Explicitly stated contributions in the introduction for clarity.
- Included new experimental results on ImageNet-100.
- Added an analysis of what is learned and ignored in representations using the synthetic dataset, with revised explanations to provide further insights.
- Included justification for using IVW-based encoders, the InfoNCE objective, and Mahalanobis distance as the critic function.

With the deadline now less than 7 hours away, we kindly remind reviewers that we remain available to promptly address any remaining concerns.\
If you feel that your concerns have been adequately resolved, we would greatly appreciate it if you could consider revisiting your review score.

Thank you,\
The Authors

---

### Meta-Review · Area_Chair_ebQT · 2024-12-15

**Metareview:**

This paper introduces a contrastive learning-based multi-view representation learning (MVRL) method designed to perform effectively even when some data sources (or "views") are missing. Experiments were conducted on seven datasets containing between 2 and 8 views. The reviewers' ratings were highly diverse, though slightly positive on average.

However, there are notable limitations. The multi-view constraint in Eq. (1) relies solely on a pairwise solution, which increases complexity as the number of views grows. Additionally, the sub-sampling mechanism lacks novelty, as multi-view data often vary significantly in perspective, with each view contributing differently. This approach may fail to capture intuitive relationships across views. Reviewer zRh5 highlighted inconsistencies in the results, such as the 3-view configuration in Table 1 performing worse than the 2-view configuration for the MOSI dataset.

**Additional Comments On Reviewer Discussion:**

The discussion did not reach a consistent output.

---

### Decision · Program_Chairs · 2025-01-22

Reject